# Noise-Guided Predicate Representation Extraction and Diffusion-Enhanced Discretization for Scene Graph Generation

Guoqing Zhang [1 2 3]   Shichao Kan [4]   Fanghui Zhang [5]   Wanru Xu [2]   Yue Zhang [6]   Yigang Cen [1 2 3]

## Abstract

Scene Graph Generation (SGG) is a fundamental task in visual understanding, aimed at providing more precise local detail comprehension for downstream applications. Existing SGG methods often overlook the diversity of predicate representations and the consistency among similar predicates when dealing with long-tail distributions. As a result, the model's decision layer fails to effectively capture details from the tail end, leading to biased predictions. To address this, we propose a **No**ise-Guided Predicate Representation Extraction and **Di**ffusion-Enhanced Di**s**cretization (NoDIS) method. On the one hand, expanding the predicate representation space enhances the model's ability to learn both common and rare predicates, thus reducing prediction bias caused by data scarcity. We propose a conditional diffusion model to reconstructs features and increase the diversity of representations for same category predicates. On the other hand, independent predicate representations in the decision phase increase the learning complexity of the decision layer, making accurate predictions more challenging. To address this issue, we introduce a discretization mapper that learns consistent representations among similar predicates, reducing the learning difficulty and decision ambiguity in the decision layer. To validate the effectiveness of our method, we integrate NoDIS

[1]State Key Laboratory of Advanced Rail Autonomous Operation, Bejing Jiaotong University, Beijing, China [2]School of Computer Science and Technology, Bejing Jiaotong University, Beijing, China [3] Visual Intelligence +X International Cooperation Joint Laboratory of MOE, Bejing Jiaotong University, Beijing, China [4]School of Computer Science and Technology, Central South University, Hunan, China [5]School of Artificial Intelligence, Henan University, Henan, China [6]College of Computer and Information Engineering, Henan Normal University, Henan, China. Correspondence to: Wanru Xu <xwanru@bjtu.edu.cn>, Yigang Cen <ygcen@bjtu.edu.cn>.

*Proceedings of the 42$^{nd}$ International Conference on Machine Learning*, Vancouver, Canada. PMLR 267, 2025. Copyright 2025 by the author(s).

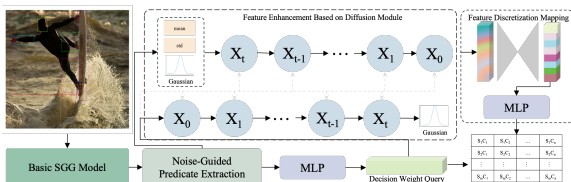

(a) NoDIS for Discretized Feature Enhancement.

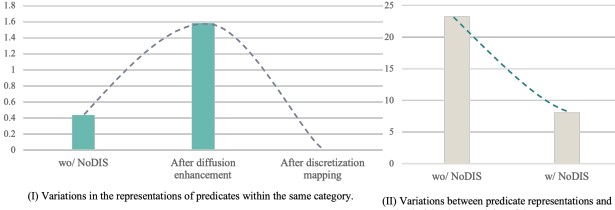

(I) Variations in the representations of predicates within the same category.

(II) Variations between predicate representations and the decision layer weights of the same category.

(b) Intra-Class Variance Changes.

*Figure 1.* (a) Our method enhances the diversity and homogeneity of representations for predicates within the same category by introducing diffusion and discretization mapping, effectively mitigating the scene graph bias problem. (b) Variance change between feature distributions. A larger variance indicates a greater difference between the feature distributions.

with various SGG baseline models and conduct experiments on multiple datasets. The results consistently demonstrate superior performance. We have uploaded the code to GitHub: https://github.com/gavin-gqzhang/NoDIS.

## 1. Introduction

Scene graph generation constructs relational representations for object pairs in an image, which provides essential insights for downstream tasks such as image captioning (Chen et al., 2020; Zhong et al., 2020), visual question answering (Hudson & Manning, 2019b; Teney et al., 2017), and image retrieval (Johnson et al., 2015; Schroeder & Tripathi, 2020), helping to understand local details within images.

However, the long-tail distribution inherent in datasets poses significant challenges. Traditional methods for predicate prediction tend to favor head predicates with shallow semantics while neglecting tail predicates with deeper meanings.

This leads to meaningless or erroneous triplet generation, severely impairing the understanding of fine-grained image details in downstream tasks like image captioning and visual question answering. To address this issue, current approaches optimize the algorithms from three perspectives: loss functions (Yan et al., 2020; Kang & Yoo, 2023), sampling strategies (Desai et al., 2021; Li et al., 2021b), and feature enhancement (Yu et al., 2023). However, these approaches concentrate exclusively on tail class enhancement while neglecting information from head classes. Recently, some methods (Li et al., 2024a; Wang et al., 2023a) have leveraged VAE (Kingma & Welling, 2022) or GAN (Goodfellow et al., 2020) for feature augmentation to strengthen the representation of tail predicates. However, these methods heavily rely on coarse-grained predicate representations as latent features and result in highly similar features before and after enhancement, which limits the effective expansion of the predicate representation space and fails to increase the diversity among similar predicates. Moreover, directly applying these methods for feature enhancement and using them to fine-tune the final decision layer weights treats each sample as an independent entity, neglecting the homogeneity among similar predicate representations, which increases the learning difficulty at the decision layer.

To address the above issues, we propose a solution from two perspectives: 1) Explore new feature enhancement methods to expand the representation space and improve the model's ability to learn generalizable representations. By increasing the diversity of intra-class predicate representations, this approach alleviates the model's underfitting on rare information, thus enabling better learning of tail-class information. 2) In the decision phase, we alleviate the learning difficulty and decision burden caused by the diverse independent representations of samples by enhancing the homogeneity among representations of predicates within the same category. By learning the unified representations of similar predicates, the decision layer is trained to learn the differences between predicate representations of different categories, thereby improving both the learning speed and decision accuracy of the decision layer.

Based on these ideas, we propose the **No**ise-Guided Predicate Representation Extraction and **Di**ffusion-Enhanced Di**s**cretization (NoDIS) method. As shown in Figure 1a, NoDIS introduces the Noise-Guided Predicate Extraction module, which extracts independent predicate features using a one-step noise addition and denoising algorithm. To expand the predicate representation space and enhance the diversity of predicate features within the same category, a conditional feature reconstruction diffusion learning method is proposed. This method significantly increases the diversity of predicate representations within the same category, leading to a notable rise in intra-class feature distribution variance (as shown in Figure 1b (I)). This enhancement

aids the model in effectively learning diverse representation knowledge. Finally, to alleviate the learning difficulty of the decision layer caused by the independence of sample representations, the Feature Discretization Mapping module is introduced. This module employs a learnable discretization encoder to learn unified representations among the same category predicates and aggregates them into an independent representation space, which is then used for learning the independent representation distributions of different categories in the decision layer. This approach effectively reduces the differences between similar predicate representations, leading to a significant decrease in intra-class feature distribution variance (as shown in Figure 1b (I)). Furthermore, by using the aggregated independent predicate representations to train the decision layer, the distribution discrepancy between similar predicate representations and their corresponding class decision weights is effectively minimized (as shown in Figure 1b (II)). This enables the decision layer to more effectively learn the aggregated independent predicate representations, thus minimizing the impact of heterogeneous predicate representations on the decision-making process and improving decision accuracy.

We conducted extensive experiments on datasets such as VG (Krishna et al., 2017), GQA (Hudson & Manning, 2019a) and OpenImage V6(Kuznetsova et al., 2020), achieving excellent performance, which demonstrates that our method effectively performs feature reconstruction and mitigates the biased predictions caused by long-tail distribution.

In summary, the contributions of this paper are as follows: 1) We propose the NoDIS method, which enhances the model's generalization ability by increasing the diversity of predicate representations within the same category. At the same time, it strengthens the homogeneity of these representations to aid the decision layer in learning the differences between representations of different categories, effectively mitigating bias prediction issues. 2) We introduce the Feature Enhancement Based on Diffusion Module, which, for the first time, applies diffusion techniques to increase the diversity of predicate representations within the same category, expand the representational space, and enhance the model's ability to generalize knowledge extraction. 3) We design a Feature Discretization Mapping module that learns consistent representations among predicates of the same category and projects them into independent representation spaces for training the decision layer. This approach effectively alleviates convergence issues and semantic ambiguity in the decision layer, thereby improving decision accuracy.

## 2. Related Works

### 2.1. Scene Graph Generation

The long-tailed distribution in datasets often leads to biased predictions in models. The early methods (Zellers et al., 2018; Xu et al., 2017; Chiou et al., 2021) aimed to mitigate this bias by taking advantage of distribution patterns within the dataset as additional prior knowledge. In recent years, one-stage Scene Graph Generation models (Cong et al., 2023) have become the hot issue of research. These models leverage Transformer-based architectures (Vaswani et al., 2017) to iteratively refine predicate representations while simultaneously determining predicates and entities. Additionally, increasing attention has been paid to the design of training strategies and loss functions(Kang & Yoo, 2023; Yan et al., 2020). Techniques such as data resampling (Li & Vasconcelos, 2019), transfer learning (Liu et al., 2019), and causal learning (Tang et al., 2019; 2020) have been explored to alleviate the impact of long-tailed distributions.

### 2.2. Diffusion Model

Diffusion has become one of the most widely used methods in generative tasks, such as image generation, super-resolution, etc. Initially (Ho et al., 2020) introduced the Denoising Diffusion Probabilistic Model (DDPM), which simulates the diffusion process and then reverses it to generate data, achieving promising results. Subsequently, Score-based Generative Models (Song et al., 2020b) were proposed, adopting a framework similar to the diffusion process but using score models to recover the data distribution in reverse. Latent Diffusion (Rombach et al., 2022) has established Diffusion as a general framework for generative tasks, significantly improving its performance and quality in image generation. Recently, researchers have increasingly focused on incorporating conditions into diffusion models (Zhang et al., 2023) to enable more precise control and higher-quality outputs.

## 3. Method

### 3.1. Problem Formulation and Symbolic Representation

We adopt a two-stage architecture for the scene graph generation task. In the first stage, Faster R-CNN (Ren et al., 2015) is used for object detection, identifying all entities in the image. Suppose that there are $i$ entities, their categories are denoted as $O = \{O_0, O_1, ..., O_i\}$, their bounding boxes as $B = \{b_0, b_1, ..., b_i\}$, and their visual features, extracted using RoIAlign (He et al., 2017), as $E = \{e_0, e_1, ..., e_i\}$. Additionally, we compute the union visual features for all entity pairs, denoted as $U = \{u_0, u_1, ..., u_j\}$, where $j = i \times (i - 1)$.

In the second stage, we focus on extracting predicate rep-

resentations and predicting predicates between entity pairs using the entity information obtained in the first stage. To achieve this, we design an independent Predicate Refinement Module. Assuming that the predicate between the $i$-th entity and the $j$-th entity is $r_{ij}$, the complete set of predicate categories can be represented as $R = \{r_{01}, r_{02}, ..., r_{ij}\}$, where $j$ corresponds to the $(i - 1)$-th entity. Additionally, we use pre-trained embeddings GloVe (Pennington et al., 2014) to encode all predicate categories, initializing the predicate prototype representations $P = \{p_1, p_2, \ldots, p_c\}$, where $c$ represents the total number of predicate categories.

### 3.2. Noise-Guided Predicate Representation Extraction

To enhance the independence of predicate representations and provide effective prior information for subsequent predicate reconstruction, we construct predicate representations from a node-edge-node structural perspective and decouple entity-predicate representations from an entity-noise perspective. As illustrated in Figure 2(a), the overall process is divided into two parts: Neighborhood Context Extraction and Noise-Guided Predicate Refinement, as follows.

**Neighborhood Context Extraction:** We treat entity pairs as nodes and construct learnable Query Tokens to represent edges between entity-pair nodes. By iteratively optimizing the learning capability of Query Tokens, we achieve an effective predicate representation construction between entities.

First, we use entity representations preprocessed by the base model (Zellers et al., 2018; Vaswani et al., 2017; Zheng et al., 2023b) to construct entity pair representations $E_s$ and $E_o$, representing subject and object entity pairs, respectively. Then, through two distinct cross-attention modules, the two entity representations alternately serve as Query, enhancing the uniform expression between entities, as shown in Eq. 2.

$$Attn(Q, K, V) = Softmax(\frac{QK^T}{\sqrt{d}})V \qquad (1)$$

$$E_s = Attn(E_s, E_o, E_o), E_o = Attn(E_o, E_s, E_s) \quad (2)$$

Next, leveraging the cross-attention modules, we employ a learnable Query Token $Q_p$ to extract consistency representations between entity pairs. This consistency representation serves as a shared edge structure for entity pairs, representing the predicate, as illustrated in Eq. 3. Additionally, to prevent information loss, we introduce a joint visual representation $U$ of the entity pairs to refine the constructed predicate representation. As shown in Eq. 4, we filter the node information of entity pairs from the joint visual representation, suppressing regions with weak correlations to the nodes of entities while strengthening the intensity of the representation of the edges between nodes.

$$Q_p = Attn(Q_p, [E_s, E_o], [E_s, E_o]) \qquad (3)$$

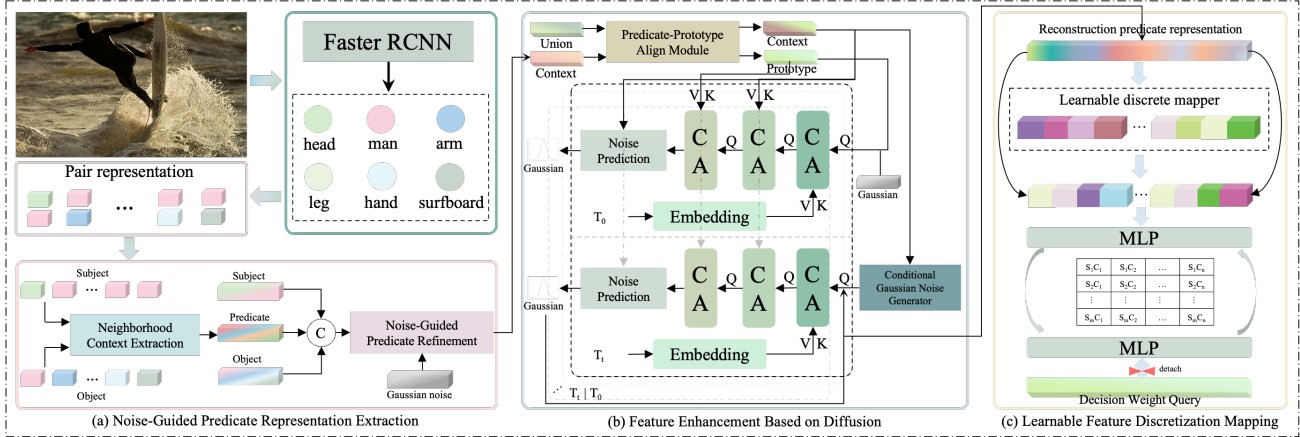

*Figure 2.* Diagram of the overall structure for the Noise-Guided Predicate Representation Extraction and Diffusion-Enhanced Discretization (NoDIS) method.

$$\begin{cases} U^{'} = Attn(U, [E_s, E_o], [E_s, E_o]), \\ U^{'} = U - \varphi(\omega([E_s, E_o])) \cdot U^{'}, \\ Q_p = Attn(Q_p, U^{'}, U^{'}) \end{cases} \quad (4)$$

In the above equations, $[,]$ denotes the concatenation operation between features. $\omega$ represents a learnable parameter matrix that is used for feature mapping and dimensional alignment. $\varphi$ stands for the Sigmoid function, which calculates feature weights and smoothly maps them to the range of 0 to 1.

**Noise-Guided Predicate Refinement:** Based on coarse-grained predicate representations extracted from the Neighborhood Context Extraction Module, we construct triplet representations $T$ by incorporating entity pair information and enhancing them using joint visual features, as shown in Eq. 5. For the initialized triplet representations, random Gaussian noise $G$ is added, with the noise intensity $\lambda$ randomly initialized and injected only once during training.

$$T = [E_s, Q_p, E_o], T = T + \omega(\varphi(\omega(T))) \cdot U \quad (5)$$

$$G \sim \mathcal{N}(0,1), T_n = T + \lambda \cdot G \quad (6)$$

In the denoising process, the goal is to use the entity pair representations within the triplet and the added noise as denoising targets to obtain clean predicate representations. To achieve this, enhanced entity representations are concatenated to construct clean entity pair representations $E$, and feature filtering is applied to remove noise effects, as described in Eq. 7. Subsequently, a cross-attention module is employed, using the noisy triplet $T_n$ as the Query and the clean entity pair $E$ as the Key and Value to strengthen the representation of entity-related regions within the noisy triplet, as shown in Eq. 8.

$$E = [E_s, E_o], E = E - \omega(\sigma(\omega(E))) \cdot E \quad (7)$$

$$T_n^{'} = Attn(T_n, E, E) \quad (8)$$

In the above equation, $\sigma$ refers to the ReLU function, which filters out irrelevant features or noise.

By enhancing the entity representation within the noisy triplet, we subtract the entity-strengthened triplet representation $T_n^{'}$ from the noisy triplet representation $T_n$, as described in Eq. 9. Since the noise in the noisy triplet is distributed across the entire representation space, this subtraction reduces the overall noise intensity. By iterating this process multiple times, both entity-related and noise components are progressively removed, ultimately yielding independent and clean predicate representations.

$$T_n = T_n - \varphi(\omega(E)) \cdot T_n^{'} \quad (9)$$

Finally, predicate representations are extracted from the denoised triplet representations and filtered, resulting in the preliminary refinement of the predicate context representation $C_p$.

$$C_p = \omega(\sigma(\omega(T_n[1]))) \quad (10)$$

### 3.3. Feature Enhancement Based on Diffusion

To expand the visual space of predicate representations and enhance feature diversity, we propose a Diffusion-based Feature Reconstruction Enhancement Module. Similar to existing diffusion models, it adheres to the same training and testing rules. However, during testing, we construct a conditional Gaussian distribution for feature reconstruction instead of using a random Gaussian distribution. As illustrated in Fig.2(b), the preprocessed predicate representations are first aligned with the prototype representations. Using a shared weight matrix, the predicate representations and prototype representations are mapped into a unified visual space, followed by refinement through joint visual

representations, as expressed in Eq.11.

$$\begin{cases} P' = MLP(P), C_p = MLP(C_p), \\ U' = Attn(U, P', P'), C_p = Attn(C_p, U', U') \end{cases}$$
(11)

During the diffusion process, we use the predicate prototypes corresponding to the ground-truth categories of all samples as the target feature distribution $T_p$. Noise of varying intensities is then added based on the current timestep, where the noise schedule follows the same strategy as (Luo & Hu, 2021). Assuming the timestep is $t$, the noise intensity is denoted as $\gamma_t$.

$$\begin{cases} T_p = P[Gt], Gt = \{Gt_1, Gt_2, ..., Gt_i\} \\ T'_p = \sqrt{\gamma_t} \cdot T_p + \sqrt{(1 - \gamma_t)} \cdot G \end{cases}$$
(12)

In the above equations, $Gt$ represents the ground truth predicate category for each sample, while $Gt_1$ and $Gt_i$ denote the predicate categories for the 1-th and i-th samples, respectively. $G$ refers to a Gaussian distribution initialized randomly, which is added as noise to the predicate prototypes corresponding to the ground truth categories of each sample.

In the noise prediction process, we introduce a conditional noise prediction module, as shown in Equation 13. Based on a Cross-Attention mechanism, the noise-added target distribution $T'_p$ is used as the Query, while the current timestep embedding, prototype representation, and predicate representation serve as the Key and Value, respectively, to progressively enhance effective feature representations. Finally, a simple linear mapping and bias learning are employed to refine the enhanced representations and predict irrelevant regions as noise, as described in Equation 14.

$$\begin{cases} E_t = Embedding(t), T'_p = Attn(T'_p, E_t, E_t), \\ T'_p = Attn(T'_p, C_p, C_p), T'_p = Attn(T'_p, T_p, T_p) \end{cases}$$
(13)

$$N' = \omega(T'_p) \cdot \varphi(\omega(C_p)) + \omega(C_p)$$
(14)

Based on the above method, noise prediction for the target feature distribution with added noise can be effectively achieved. However, during the reconstruction phase, Gaussian noise with random initialization, which contains no prior information, is used as input and gradually denoised. This process may result in reconstructed features deviating from the target feature distribution. To address this issue, we introduce a parallel branch during training for feature reconstruction, further constraining the feature generation direction. Specifically, Gaussian noise distributions are reconstructed using the mean and variance of the preprocessed predicate representations $C_p$. Reverse diffusion is then applied to predict the noise and iteratively denoise the features. By randomly selecting $n$ time steps, noise is predicted iteratively and denoised according to the noise intensity $\gamma_t$,

yielding the reconstructed predicate representations $G_p$, as shown in Algorithm 1. During testing, all time steps are used instead of selecting $n$.

---

**Algorithm 1** Feature Reconstruction Training Process Based on Diffusion

---

1: $T, n$ {T: total number of iteration steps, n: randomly selected step sizes}
2: $m = \omega(C_p), v = \omega(C_p)$ {Init feature distribution}
3: $G' \leftarrow m + \sqrt{v} \cdot \mathcal{N}(0, 1)$ {Init noise input}
4: $G_p \leftarrow G'$
5: **for** t in $random(T, n)$ **do**
6:    $E_t \leftarrow Embedding(t)$ {Initialize time embedding}
7:    $N' \leftarrow Attn(G_p, E_t, E_t)$ {Conditional diffusion}
8:    $N' \leftarrow Attn(N', C_p, C_p)$ {Conditional diffusion}
9:    $N' \leftarrow Attn(N', T_p, T_p)$ {Conditional diffusion}
10:   $N' \leftarrow \omega(N') \cdot \varphi(\omega(C_p)) + \omega(C_p)$ {Noise prediction}
11:   $G_p \leftarrow \sqrt{\gamma_{t-1}} \cdot \left( \frac{G' - \sqrt{1-\gamma_t} \cdot N'}{\sqrt{\gamma_t}} \right)$ {Single-step denoising}
12: **end for**
13: **if** is training **then**
14:   $loss \leftarrow MSELoss(G_p, T_p)$
15:   **return** $G_p, loss$
16: **end if**
17: **return** $G_p$

---

### 3.4. Learnable Feature Discretization Mapping Module

We observe that using Diffusion for predicate representation reconstruction can lead to a discrete distribution where each sample corresponds to a unique feature, which hinders subsequent predicate classification. To address this, we propose the introduction of a discrete encoder similar to that in (Van Den Oord et al., 2017). First, we construct a learnable parameter matrix $L_p$ and initialize it using preprocessed encoded predicate prototypes $P'$. Next, we calculate the distance between the reconstructed predicate representations and the features of each category in $L_p$. The closest parameter feature to the reconstructed feature is selected as the affine-transformed feature $G_l$, as shown in Eq. 15.

$$\begin{cases} L_p \sim P', Dis = \sqrt{\sum(G_p - L_p)^2}, \\ G_l = L_p[argmin(Dis)] \end{cases}$$
(15)

In the above equation, $G_p$ represents the predicate representations reconstructed based on Algorithm 1, and $Dis$ represents the distance between the reconstructed features and the discretized prototype features. Assuming the number of samples is $m$ and the number of predicate categories is $c$, the scale of $Dis$ is $(m, c)$. $argmin(Dis)$ denotes the index of the shortest distance between each reconstructed sample and the discretized representation.

Finally, we perform feature mapping on the affine-transformed feature $G_l$ and aligned predicate prototypes $P^{'}$ from Eq. 11, and compute the similarity between the discretized affine-transformed features and prototype representations. This similarity serves as the final prediction of the predicate category, as shown in Eq. 16. During training, the pre-processed aligned predicate prototypes $P^{'}$ are not affected by the gradients from this part, and the computation graph is disconnected, allowing independent updates for optimization.

$$\begin{cases} G_l = MLP(G_l), P_g = MLP(\beth(P^{'})), \\ R_d = G_l \times P_g \end{cases} \quad (16)$$

where $\beth$ represents the stopping of gradient propagation.

### 3.5. Loss Function

**Noise-Guided Predicate Representation Extraction:** In this module, our loss function consists of three main components. we adopt the predicate representation and prototype distance metric loss proposed in PENet (Zheng et al., 2023b) to jointly constrain the predicate representations $Q_p$ and $C_p$ extracted from two subcomponents. The Euclidean distance is used to compute the distances between $Q_p$, $C_p$, and the predicate prototypes, resulting in the losses $L_{Qd}$ and $L_{cd}$. These losses enforce minimal intra-class predicate representation distances and maximal inter-class predicate representation distances.

Additionally, we introduce a re-weighted cross-entropy loss function, where the weights are dynamically adjusted based on the prediction status. Specifically, we construct positive and negative prediction score tables, $S_p$ and $S_N$, respectively. The prediction score for the true category of each sample is used as the positive sample weight, while the prediction scores for other categories serve as negative samples. We also track the occurrence frequency of each sample across all categories, denoted as $S_t$. The dimensions of $S_p$, $S_N$ and $S_t$ are all $(c, )$, where $c$ is the number of categories. Then, using a momentum update method, the loss weights for each category are updated iteratively. The initial weight for each category $S_i$ is set to 1.

$$\begin{cases} R_c = C_p \times P^{'}, S_t = S_t + P_{gt}, \\ S_p = S_p + R_c \cdot P_{gt}, S_N = S_N + R_c \cdot \overline{P_{gt}}, \\ S = S_i \cdot \delta + (1-\delta) \cdot \lg \left(1 + \frac{\frac{S_N}{S_t+\xi}}{\frac{S_p}{S_t+\xi}+\xi}\right), \\ L_{ace} = CE(R_c, Gt, weight = S) \end{cases} \quad (17)$$

In the above equation, $P_{gt}$ represents the one-hot encoded distribution of the true predicate categories, where the index of the true category for each sample is set to 1. $\overline{P_{gt}}$ denotes the inverse of the one-hot encoded distribution of the true predicate categories.

Finally, we incorporate a KL divergence-based feature constraint. By calculating the feature distribution between the extracted predicate representations and the predicate prototypes, this constraint optimizes the distribution of the extracted representations to align with the feature distribution of the prototypes.

$$\begin{cases} m_c = \omega(C_p), m_p = \omega(P^{'}), \\ logv_c = \omega(C_p), logv_p = \omega(P^{'}), \\ L_{kl} = -\frac{1}{2}\sum(1 - \frac{e^{logv_c}}{e^{logv_p}} - \frac{(m_c-m_p)^2}{e^{logv_p}} + e^{logv_c} - e^{logv_p}) \end{cases} \quad (18)$$

In the above equations, $m_c$ and $m_p$ share the same weight matrix, $logv_c$ and $logv_p$ share the same weight matrix.

**Feature Enhancement Based on Diffusion:** First, to enhance the noise prediction capability of the model, we use MSE loss to calculate the loss between the predicted noise $N^{'}$ (from Eq. 14) and the initial added noise $G$ (from Eq. 12), thereby optimizing the noise extraction process. Second, to effectively control the denoising direction, we follow the computation steps in Algorithm 1, randomly selecting time steps for noise prediction and iterative denoising. Finally, the denoised predicate representations $G_p$ is compared with the predicate prototype $T_p$ corresponding to the sample's category (from Eq. 12) by the MSE loss to ensure feature consistency between the reconstructed representation and the prototype.

$$L_{rec} = ||N^{'} - G||^2 + ||G_p - T_p||^2 \quad (19)$$

**Learnable Feature Discretization Mapping Module:** For discretizing the reconstructed predicate representations, we adopt the loss function design of (Van Den Oord et al., 2017) to calculate the quantization loss between features before and after mapping. Additionally, we supervise the distance between features to prevent the reconstructed representations from being associated with incorrect discrete encoded features. As shown in Eq. 20, the distance $Dis$ (defined in Eq. 15) between the reconstructed representation and the discrete encoded representation is normalized and constrained by the ground truth labels of the samples. This ensures that the reconstructed representations remain homogeneous with the discrete encoded representations of the same category.

$$L_{dis} = ||G_p - \beth(G_l)||^2 + ||\beth(G_p) - G_l||^2 + ||Dis - P_{gt}||^2 \quad (20)$$

## 4. Experiments

### 4.1. Experiment Setting

**Dataset:** We use the Visual Genome (VG) (Krishna et al., 2017) and GQA (Hudson & Manning, 2019a) datasets for

| Method | PredCls | | | SGCls | | | SGDet | | |
|---|---|---|---|---|---|---|---|---|---|
| | mR@50/100 | R@50/100 | F@50/100 | mR@50/100 | R@50/100 | F@50/100 | mR@50/100 | R@50/100 | F@50/100 |
| RelTR (Cong et al., 2023) TPAMI'23 | 21.2/- | **64.2**/- | 31.9/- | 11.4/- | 36.6/- | 17.4/ | 10.8/- | 27.5/- | 15.5/- |
| HetSGG (Yoon et al., 2023) AAAI'23 | 32.3/34.5 | 57.1/**59.4** | 41.3/43.6 | 15.8/17.7 | **37.6/38.5** | 22.3/24.3 | 11.5/13.5 | **30.2/34.5** | 16.7/19.4 |
| DCNet (Han et al., 2022a) TCSVT'22 | 33.4/35.6 | 57.3/59.1 | **42.2/44.4** | **21.2**/22.2 | 36.0/36.8 | **26.7/27.7** | **14.3**/17.3 | 28.6/32.9 | **19.1**/22.7 |
| ST-SGG (Kim et al., 2024) ICLR'24 | 32.7/35.6 | 52.5/54.3 | 40.3 / 43.0 | 21.0/**22.4** | 36.3/37.3 | 26.6 / **27.9** | 12.6/15.1 | 20.7/24.9 | 15.7 / 18.8 |
| PCPL (Yan et al., 2020) MM'20 | 35.2/37.8 | 50.8/52.6 | 41.6/44.0 | 18.6/19.6 | 27.6/28.4 | 22.2/23.2 | 9.5/11.7 | 14.6/18.6 | 11.5/14.4 |
| HiKER (Zhang et al., 2024) CVPR'24 | **39.3/41.2** | -/- | -/- | 20.3/21.4 | -/- | -/- | -/- | -/- | -/- |
| EGTR (Im et al., 2024) CVPR'24 | -/- | -/- | -/- | -/- | -/- | -/- | 14.0/**18.3** | 28.2/31.7 | 18.7/**23.2** |
| VTransE (Zhang et al., 2017) CVPR'17 | 17.4/18.7 | **66.0/67.8** | 27.5/29.3 | 11.2/12.3 | **40.3/41.4** | 17.5/19.0 | 7.3/8.5 | **31.4/35.6** | 11.8/13.7 |
| GCL (Dong et al., 2022) CVPR'22 | 34.2/36.3 | -/- | -/- | 20.5/21.2 | -/- | -/- | 13.6/15.5 | -/- | -/- |
| +NoDIS(ours) | **35.68/38.04** | 57.83/ 59.74 | **44.13/46.48** | 20.83/22.14 | 38.84/39.96 | **27.12/28.49** | 13.77/15.83 | 27.96/31.97 | **18.45/21.18** |
| Motifs (Zellers et al., 2018) CVPR'18 | 17.4/19.3 | 65.7/67.9 | 27.5/30.0 | 10.9/12.0 | **41.3**/42.5 | 17.3/18.8 | 7.3/8.6 | 32.0/36.3 | 11.9/14.0 |
| +DBiased (Han et al., 2022b) TM'22 | 34.7/36.6 | 58.8/60.7 | 43.6/45.7 | 20.3/21.2 | 36.5/37.4 | 26.1/27.1 | **14.9/ 17.5** | 29.4/33.9 | **19.8/ 23.1** |
| +PCL (Tao et al., 2022) TIP'22 | 33.6/35.8 | 55.0/57.3 | 41.7/44.1 | 18.2/19.1 | 34.2/35.2 | 23.8/24.8 | 14.2/16.6 | 29.0/33.4 | 19.1/22.2 |
| +NICE (Li et al., 2022a) CVPR'22 | 29.9/32.3 | 55.1/57.2 | 38.8/41.3 | 16.6/17.9 | 33.1/34.0 | 22.1/23.5 | 12.2/14.4 | 27.8/31.8 | 17.0/19.8 |
| +HML (Deng et al., 2022) ECCV'22 | **36.3/ 38.7** | 47.1/49.1/ | 41.0/43.3 | 20.8/22.1 | 26.1/27.4 | 23.2/24.5 | 14.6/ 17.3 | 17.6/21.0 | 16.0/19.0 |
| +Inf (Biswas & Ji, 2023) CVPR'23 | 24.7/30.7 | 51.5/55.1 | 33.4/39.4 | 14.5/17.4 | 32.2/33.8 | 20.0/23.0 | 9.4/11.7 | 23.9/27.1 | 13.5/16.3 |
| +LS-KD (Li et al., 2023b) TCSVT'23 | 34.2/37.9 | 47.6/50.9 | 39.8/43.4 | 18.7/20.9 | 31.2/32.7 | 23.4/25.5 | 13.7/16.6 | 23.5/27.2 | 17.3/20.6 |
| +QuatRE (Wang et al., 2023c) TMM'23 | 15.8/17.1 | 66.2/ 68.0 | 25.5/27.3 | 9.2/9.8 | 40.3/41.2 | 15.0/15.8 | 6.2/7.6 | **32.5/ 37.4** | 10.4/12.6 |
| +CFA (Li et al., 2023a) ICCV'23 | 35.7/ 38.2 | 54.1/56.6 | 43.01/45.61 | 17.0/18.4 | 34.9/36.1 | 22.86/24.38 | 13.2/15.5 | 27.4 31.8 | 17.82/20.84 |
| +MEET (Sudhakaran et al., 2023) ICCV'23 | 25.3/33.5 | **67.4/ 72.7** | 36.79/ 45.87 | 19.0/ 23.7 | 40.5/ **43.2** | 25.87/ **30.61** | 8.5/11.8 | 27.9/33.3 | 13.03/17.43 |
| +NoDIS(ours) | 33.83/36.48 | 62.72/64.98 | **43.95/46.73** | 22.68/24.35 | 36.46/37.48 | 27.96/29.52 | 13.58/16.35 | 30.91/35.42 | 18.87/22.37 |
| Transformer (Vaswani et al., 2017) NIPS'17 | 21.4/23.7 | **66.7/68.8** | 32.4/35.2 | 12.2/13.3 | **41.7/ 42.7** | 18.9/20.3 | 7.6/9.1 | **31.6/ 35.9** | 12.3/14.5 |
| +CogTree (Yu et al., 2020) IJCAI'21 | 28.4/31.0 | 38.4/39.7 | 32.7/34.8 | 15.7/16.7 | 22.9/23.4 | 18.6/19.5 | 11.1/12.7 | 19.5/21.7 | 14.1/16.0 |
| +HML (Deng et al., 2022) ECCV'22 | 33.3/ 35.9 | 45.6/47.8 | 38.5/ 41.0 | 19.1/ 20.4 | 22.5/23.8 | 20.7/ 22.0 | 15.0/ 17.7 | 15.4/18.6 | 15.2/ 18.1 |
| +ALF (Chen et al., 2024) MM'24 | 35.9/38.4 | 53.3/55.2 | **42.9**/45.3 | 21.3/22.6 | 29.0/30.0 | 24.6/25.8 | 14.8/17.5 | 22.8/26.2 | 17.9/21.0 |
| +BiC (Yang et al., 2024) TIP'24 | 34.6/ 37.2 | 53.4/56.0 | 42.0/44.7 | 19.5/ 21.0 | 33.0/34.0 | 24.7/26.0 | 16.7/ 19.1 | 23.3/27.3 | 19.5/22.5 |
| +NoDIS(ours) | **37.25/39.97** | 49.96/53.21 | 42.68/**45.65** | 21.53/23.35 | 34.42/35.63 | 26.49/28.21 | 16.91/19.25 | 25.26/28.77 | 20.26/23.07 |
| PE-Net (Zheng et al., 2023b) CVPR'23 | 31.5/33.8 | **64.9/ 67.2** | 42.4/45.0 | 17.8/18.9 | **39.4/ 40.7** | 24.5/25.8 | 12.4/14.5 | **30.7/ 35.2** | 17.7/20.5 |
| +NoDIS(ours) | **38.72/41.93** | 50.13/53.87 | **43.69/47.16** | 22.31/23.69 | 30.65/32.19 | 25.82/27.29 | 14.78/17.16 | 23.16/26.95 | 18.05/20.97 |

*Table 1.* In the VG dataset, we compare the performance of our method with existing methods in the PredCls, SGCls, and SGDet tasks based on R@K, mR@K and F@K (%) metrics.

model training and evaluation. The VG dataset consists of 108,077 images, and following the decomposition method in (Xu et al., 2017), we created a dataset containing 150 object categories and 50 relationship categories. Compared to VG, the GQA dataset GQA dataset includes 200 object categories and 100 relationship categories. Both VG and GQA datasets are split using the same method: 70% of the samples are used for training, 30% for testing, with 5,000 samples selected from the training set for validation.

Additionally, we employed the Open Images (Kuznetsova et al., 2020) dataset to further evaluate the generalization capability of our method. Open Images contains approximately 9 million images, 600 object categories, and 375k annotated relationships. Open Image V6 extends this to 391k visual relationships and introduces action annotations and localized narratives, enhancing the diversity of localized understanding. Following the data processing approach proposed in (Li et al., 2021a; Zheng et al., 2023b; Lin et al., 2020; Zhang et al., 2019), we used 126,368 images for training, 1,813 for validation, and 5,322 for testing.

**Evaluation Tasks and Metrics:** We follow traditional task evaluation methods (Tang et al., 2019) , training and evaluating on three tasks: PredCls (Predicate Classification), SGCls (Scene Graph Classification), and SGDet (Scene Graph Detection). For performance evaluation, we use R@K, mR@K, and F@K (Zhang et al., 2022) metrics, with K set to 20, 50, and 100.

### 4.2. Comparison with state-of-the-art Methods

We trained and evaluated three tasks on the VG dataset (Krishna et al., 2017). Considering that additional prior information or augmentation methods are not available in real-world scenarios, we did not use any data augmentation or prior knowledge, relying solely on the existing data and model structure. As shown in Table 1, our method significantly improves the mR@K metric, outperforming the current state-of-the-art methods. However, our method may not be optimal in terms of R@K, as mR@K evaluates overall accuracy across the head, body, and tail categories, while R@K focuses more on the performance of local categories. Additionally, since the Motifs model uses prior knowledge during training, removing this prior information affected its representation ability, leading to suboptimal performance. As illustrated in Figure 3, our method effectively enhances the basic model's prediction performance for body and tail categories while maintaining its accuracy for the head category, ultimately improving overall prediction accuracy.

Meanwhile, based on the SGDet experimental setting, we also conducted experiments on the Open Images V4 and V6 (Kuznetsova et al., 2020) datasets, as shown in Table 2. We adopted a simple Transformer (Vaswani et al., 2017) as our baseline model. Compared to VCTree (Tang et al., 2019) and Motifs (Zellers et al., 2018), the Transformer has relatively weaker feature extraction capabilities and fewer parameters, making it less complex overall. However, when combined with our proposed method, its discriminative

ability on these datasets is significantly improved, achieving superior performance across multiple metrics. Specifically, the $wmAP_{rel}$ and $wmAP_{phr}$ metrics intuitively reflect the model's ability to detect predicate categories and complete triplet phrases, respectively. As shown in Table 2, our method consistently achieves the best performance in $wmAP_{rel}$, $wmAP_{phr}$, and $score_{wtd}$ on both the Open Images V4 and V6 datasets. This demonstrates that our approach not only offers strong generalization capability but also accurately identifies triplet information.

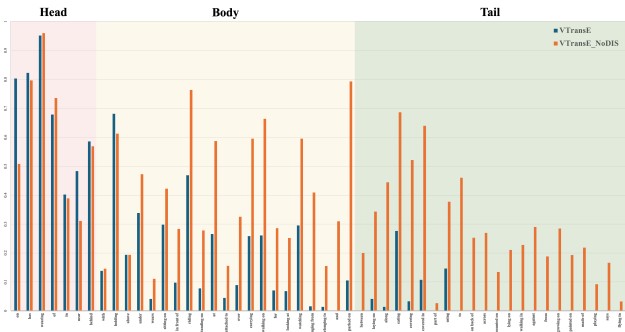

*Figure 3.* Recall values for each category with VTransE (Zhang et al., 2017) and after incorporating NoDIS, with predicate frequencies decreasing from left to right in the dataset.

We further analyze and compare the model parameters, as shown in Fig. 4. Specifically, we integrate NoDIS with three representative baseline models: Motifs (Zellers et al., 2018), VCTree (Tang et al., 2019), and Transformer (Vaswani et al., 2017), and measure the change in parameter counts. Although incorporating our method leads to an increase of approximately 40–60M parameters, this growth is relatively modest compared to other widely adopted methods or structures (Tang et al., 2020; Suhail et al., 2021; Li et al., 2021a). Moreover, despite the smaller increase in parameters, our method yields more significant improvements in model performance. Detailed evaluation and performance comparison analysis can be found in Appendix B.

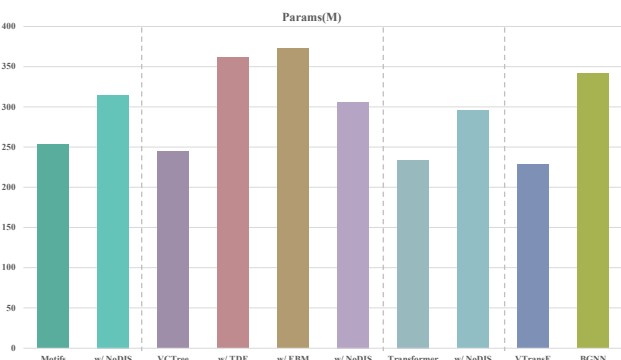

*Figure 4.* Comparison of Parameter Counts After Integrating NoDIS with Base Models (Zellers et al., 2018; Tang et al., 2019; Vaswani et al., 2017).

### 4.3. Ablation Studies

We conducted ablation studies from four perspectives: predicate representation extraction, loss function design, diffusion-based feature enhancement, and discretized feature aggregation. As shown in Table 3, using only noise-guided decoupled predicate representation or predicate prototype alignment refinement results in negligible performance improvement. However, when both modules are introduced simultaneously, the performance on mR@100 improves by at least 2.25%. For the loss function, as illustrated in Table 4, incorporating only adaptive re-weighted cross-entropy loss or KL loss individually yields limited performance gains. When both adaptive re-weighted cross-entropy loss and KL loss are jointly applied to constrain predicate representations, the performance significantly improves, with an increase of at least 2.43% on mR@100.

Finally, regarding diffusion-based feature enhancement and discretized feature aggregation, as shown in Table 5, introducing either module alone does not lead to noticeable performance improvements and may even result in performance degradation. However, when both modules are applied together for feature enhancement and aggregation, the performance improves significantly, achieving a 1.34% increase compared to the baseline without these modules. Detailed ablation analysis can be found in Appendix C.

### 4.4. Quantitative and Qualitative Analysis

**Quantitative Analysis** Based on the distribution variance of predicate representations within the same category, as shown in the Figure 1b (I), after applying the Diffusion method for feature enhancement, the variance of features within the same category increases. This indicates that the introduction of the Diffusion method effectively expands the feature space, enhancing the diversity of predicate representations within the same category. After applying the discretization mapping module, the variance between predicate representations sharply decreases. This suggests that discretization mapping effectively learns consistent representations from the expanded feature space, strengthening the homogeneity among predicate representations within the same category.

Analyzing the distribution between predicate representations and decision head weights, as shown in the Figure 1b (II), after enhancing the homogeneity of similar predicate representations, the distributions of predicate representations and decision weights within the same category become more similar. Without NoDIS method, the feature distribution exhibits greater variance. This indicates that the introduction of the NoDIS module effectively enhances the homogeneity of similar predicate representations, which aids the decision layer in making accurate and efficient category predictions. For further quantitative analysis, please refer to Appendix

| Dataset | Method | mR@50 | R@50 | F@50 | wmAP_rel | wmAP_phr | score_wtd |
|---|---|---|---|---|---|---|---|
| OI V4 | RelDN (Zhang et al., 2019) | 70.4 | **75.7** | 73.0 | 36.1 | 39.9 | 45.2 |
| | GPS-Net (Lin et al., 2020) | 69.5 | 74.7 | 72.0 | 35.0 | 39.4 | 44.7 |
| | BGNN (Li et al., 2021a) | **72.1** | 75.5 | **73.8** | 37.8 | 41.7 | 46.9 |
| | **Transformer-NoDIS(Ours)** | 70.34 | 74.84 | 72.52 | **38.21** | **42.35** | **47.19** |
| OI V6 | DBiased (Han et al., 2022b) | 42.1 | 74.6 | 53.8 | 34.3 | 34.4 | 42.3 |
| | PGSG (Li et al., 2024b) | 40.7 | 62.0 | 49.1 | 19.7 | 27.8 | 28.7 |
| | HOTR (Kim et al., 2021) | 40.1 | 52.7 | 45.5 | 19.4 | 21.5 | 26.9 |
| | PE-Net (Zheng et al., 2023b) | 39.3 | **76.5** | 51.9 | 36.6 | 37.4 | 44.9 |
| | SGTR (Li et al., 2022b) | 42.6 | 59.9 | 49.8 | 38.7 | 37.0 | 42.3 |
| | CSL (Liu et al., 2023b) | 41.7 | 75.4 | 53.7 | 34.3 | 35.4 | 42.9 |
| | BCTR (Hao et al., 2025) | 48.8 | 68.6 | 57.0 | 36.0 | **39.0** | 43.7 |
| | SQUAT (Jung et al., 2023) | - | 75.8 | - | 34.9 | 35.9 | 43.5 |
| | **Transformer-NoDIS(Ours)** | **48.93** | 74.11 | **58.94** | **38.87** | **38.95** | **45.95** |

*Table 2.* We compared with the state-of-the-art methods on Open Image(Kuznetsova et al., 2020) dataset using the evaluation metrics proposed in (Li et al., 2021a).

| NPR | PPA | R@50/mR@50 | R@100/mR@100 |
|---|---|---|---|
| ✓ | ✗ | **59.89**/31.76 | **62.32**/34.06 |
| ✗ | ✓ | 47.42/33.19 | 49.61/36.16 |
| ✓ | ✓ | 54.26/**35.76** | 56.63/**38.41** |

*Table 3.* Ablation Study on Submodules of the Noise-Guided Predicate Representation Extraction Module. NPR refers to Noise-Guided Predicate Refinement, and PPA denotes the Predicate-Prototype Align module.

| $L_{ace}$ | $L_{kl}$ | R@50/mR@50 | R@100/mR@100 |
|---|---|---|---|
| ✗ | ✗ | 59.46/31.85 | 61.89/34.22 |
| ✓ | ✗ | 59.18/33.35 | 61.60/35.98 |
| ✗ | ✓ | **59.84**/32.15 | **62.24**/34.40 |
| ✓ | ✓ | 54.26/**35.76** | 56.63/**38.41** |

*Table 4.* Ablation Study on the Loss Functions in the Noise-Guided Predicate Representation Extraction Module.

D.

**Qualitative Analysis** We conducted a qualitative evaluation of model outputs on the test dataset based on the PredCls task. As shown in Figure 5, compared to the baseline model, our method generates more accurate relationships with richer semantic information, such as "letter **painted on** train" and "person **looking at** train." Additional qualitative comparisons and analyses are provided in Appendix D.

## 5. Conclusion

This paper proposes the NoDIS method to address the bias in prediction from two perspectives: enhancing the diversity of predicates within the same class and ensuring homogeneity among similar predicate representations. First, by

| FED | Random | Condition | CGNG | FDM | R@50/mR@50 | R@100/mR@100 |
|---|---|---|---|---|---|---|
| ✗ | ✗ | ✗ | ✗ | ▲ | 42.08/34.75 | 45.34/37.99 |
| ✓ | ✗ | ✗ | ✗ | ✗ | **54.23**/35.75 | **56.60**/38.42 |
| ✓ | ✓ | ✗ | ✗ | ✗ | 45.11/36.04 | 47.95/38.95 |
| ✓ | ✓ | ✗ | ✗ | ▲ | 45.34/36.03 | 48.13/39.18 |
| ✓ | ✓ | ✓ | ✗ | ▲ | 45.51/36.18 | 48.31/39.27 |
| ✓ | ✓ | ✓ | ✓ | ▲ | 49.98/**36.90** | 52.44/39.66 |
| ✓ | ✓ | ✓ | ✓ | ▼ | 49.72/36.85 | 52.21/**39.75** |

*Table 5.* Ablation Study of the Feature Enhancement Based on Diffusion (FED) Module and Feature Discretization Mapping (FDM) Module. "Random" indicates feature reconstruction with random time steps, "Condition" introduces the conditional diffusion method, ▲ denotes providing prior conditions for Diffusion, and ▼ indicates discretizing the features after Diffusion-based reconstruction.

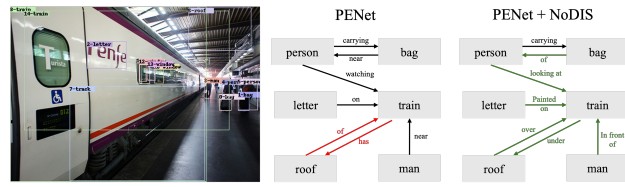

*Figure 5.* Qualitative comparisons are conducted on the test dataset based on the PredCls task. Black arrows and descriptions indicate correct predicate relationships, red ones represent incorrect predicate relationships, and green ones signify more accurate and superior predicate relationships.

introducing a conditional diffusion model, NoDIS expands the predicate representation space, enabling the model to better learn underrepresented knowledge from tail classes. Second, a discretization encoder is designed to seek consistent representations among predicates of the same class and aggregate them into an independent representation space for training at the decision layer, thereby alleviating issues of insufficient learning and decision confusion caused by the discrete nature of independent samples.

## Acknowledgements

This work was supported in part by the National Natural Science Foundation of China under Grant 62473033, 62463002, 62406105 and 62202499, in part by the Beijing Municipal Natural Science Foundation, China under Grant L231012, in part by the Science and Technology Research Project of Henan Province 242102210055 and 252102211026.

## Impact Statement

The proposed work addresses the issue of biased predictions in scene graph generation tasks. Errors in determining relationships between objects may lead to misjudgments, potentially resulting in incorrect understanding of local image details. However, since the relationship categories are constrained during the determination process, the risk of potential ethical concerns is effectively avoided.

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

## A. Implementation Details

We use a pre-trained Faster RCNN (Tang et al., 2020; Ren et al., 2015) for object detection, with the detector frozen during all three tasks. The training process is divided into two phases. First, the basic scene graph generation model (Zellers et al., 2018; Vaswani et al., 2017; Tang et al., 2019) provides coarse-grained contextual information, which is used for pretraining the Noise-Guided Predicate Representation Extraction module. This is done by leveraging the loss function of the module to constrain the contextual representation of predicates. Next, based on the extracted predicate context representations, we begin to train the two feature enhancement modules. During training, the learning rate is set to 0.001. In the pre-training phase, the batch size is set to 8, and the number of iterations is 60,000. In the feature enhancement phase, the batch size is set to 8, and the number of iterations is 40,000. All experiments are conducted using four NVIDIA 3090 GPUs, each with 24GB of memory.

## B. Experimental Analysis

We first integrate NoDIS with existing popular SGG models (Zellers et al., 2018; Zhang et al., 2017; Vaswani et al., 2017; Zheng et al., 2023b) and conduct extensive experiments on three tasks from the VG (Krishna et al., 2017) dataset to validate the effectiveness of our method. Existing approaches often use the frequency of different triplets in the dataset as additional prior knowledge, which is weighted against the model's final prediction scores. However, we recognize that in real-world scenarios or with unknown data, such prior information may not be available, potentially affecting the model's performance. Therefore, we do not introduce any prior knowledge or data augmentation techniques during model training or evaluation, aiming to achieve the most reliable results with the simplest approach.

As shown in Table 1, compared to the best existing methods, our approach achieves relatively better results on mR@K metrics, indicating that our method strikes a better balance in predicting both head and tail predicates, rather than favoring only the head classes. However, our method may show lower R@K scores, as R@K reflects local prediction accuracy, while mR@K represents overall prediction performance. When the model favors head classes, it leads to improved accuracy for head class predictions, resulting in a higher R@K. However, this significantly weakens the model's ability to predict tail information. As shown in Figure 6, we can clearly observe that our method significantly improves the prediction accuracy for body and tail classes compared to the baseline model, while still maintaining the baseline model's ability to predict head classes.

Additionally, for the SGDet task, we compute Recall@100 metrics for head, body, and tail classes. As shown in Table 6,

existing models tend to have higher accuracy for head class predictions, as they are biased toward head categories, overlooking the tail categories with deeper semantic information. By incorporating NoDIS into the existing baseline model, we significantly improve the model's prediction accuracy for the body and tail category, while maintaining its prediction performance for the head class. As a result, the model achieves optimal performance in overall mean evaluation. This demonstrates that our approach effectively mitigates biased predictions caused by long-tail distributions, while enhancing the model's prediction accuracy for the body and tail predicate categories at minimal cost.

| Models | Head | Body | Tail | Mean |
|---|---|---|---|---|
| G-RCNN | 28.6 | 6.5 | 0.1 | 11.7 |
| GPS-Net (Lin et al., 2020) | 30.4 | 8.5 | 3.8 | 14.2 |
| NBP (Liu et al., 2023a) | 31.7 | 15.0 | 8.9 | 18.5 |
| BGNN (Li et al., 2021a) | 33.4 | 13.4 | 6.4 | 17.7 |
| GSL (Liu et al., 2023c) | 33.6 | 13.5 | 8.8 | 18.6 |
| RelDN (Zhang et al., 2019) | 34.1 | 6.6 | 1.1 | 13.9 |
| VTransE (Zhang et al., 2017) | 34.5 | 7.6 | 1.1 | 14.4 |
| Motifs (Zellers et al., 2018) | 34.2 | 8.6 | 2.1 | 15.0 |
| MSDN (Li et al., 2017) | **35.1** | 5.5 | 0.0 | 13.5 |
| VCTrree-TDE(Tang et al., 2020) | 24.5 | 13.9 | 0.1 | 12.8 |
| **VTransE-NoDIS (ours)** | 30.05 | 15.13 | 10.84 | 18.67 |
| **Transformer-NoDIS (ours)** | 28.35 | 19.23 | **15.16** | **20.91** |
| **PENet-NoDIS (ours)** | 27.83 | **19.87** | 11.03 | 19.58 |

*Table 6.* Under the SGDet configuration, we divided the long-tail distribution predicate categories in the VG dataset into three groups: head, body, and tail, and calculated the R@100(%) metric for each group.

Finally, we conducted experiments on the GQA (Hudson & Manning, 2019a) dataset across three tasks, as shown in Table 7. Compared to traditional methods, our approach achieves strong performance on all three subtasks. However, we acknowledge that it does not surpass the most recent state-of-the-art methods (Wang et al., 2023b; Biswas & Ji, 2023) on this dataset in terms of evaluation metrics. Nevertheless, considering overall performance, our method demonstrates consistently strong results on both the VG and Open Images datasets, and achieves relatively competitive performance on the GQA dataset as well. This indicates that our approach is effective across different scenarios and remains robust under varying scene complexities, highlighting its strong generalization capability.

## C. Ablation Studies

We conduct ablation studies from two perspectives. First, we independently analyze the components and loss functions of the Noise-Guided Predicate Representation Extraction module. Then, we separately examine the effects of the Feature Enhancement Based on Diffusion module and the Learnable Feature Discretization Mapping module. Our ablation experiments are conducted based on the Transformer framework (Vaswani et al., 2017), with an ablation analysis

performed for each module of NoDIS.

### C.1. Ablation Study of the Noise-Guided Predicate Representation Extraction Module

First, we perform an ablation analysis of the internal structure of this module. The overall structure is divided into three components: Neighborhood Context Extraction, Noise-Guided Predicate Refinement, and Predicate-Prototype Align Module. The Neighborhood Context Extraction serves as the foundational predicate context extraction module, providing basic predicate context representations and thus cannot be removed. As shown in Table 8, when the Predicate-Prototype Align module is excluded, the overall performance remains low regardless of whether the Noise-Guided Predicate Refinement module is used. However, introducing the Predicate-Prototype Align module leads to varying degrees of performance improvement, as demonstrated in Rows 3 and 4 of Table 8. This indicates that the Predicate-Prototype Align module effectively aligns the extracted predicate context representations with predicate prototypes, significantly enhancing the discriminative ability of predicates.

Section 3.5 provides a detailed explanation of the loss functions used in the Noise-Guided Predicate Representation Extraction Module. The basic loss functions $L_{Qd}$ and $L_{cd}$ serve to constrain predicate representations. When neither the Noise-Guided Predicate Refinement Module nor the Predicate-Prototype Align Module is introduced, these basic loss functions alone are used to constrain predicate extraction. Therefore, our ablation study focuses on the additional dynamic weighting loss function $L_{ace}$ and the KL divergence loss function $L_{kl}$, which constrains feature distributions.

As shown in Table 9, when only the base loss functions are used, the model performs relatively poorly. Introducing the dynamic weighting loss, as seen in the second row, significantly improves performance. When both loss functions are applied, the model achieves optimal performance. These findings indicate that while adding either loss function individually provides slight performance gains, their effects differ. The dynamic weighting loss focuses on the overall prediction state, resulting in a noticeable improvement in the mR metric. In contrast, the KL divergence loss targets local feature distribution alignment, which may still be influenced by long-tail distributions, leading to biased predictions.

### C.2. Ablation Study on Diffusion Steps and Random Sampling Steps in the Feature Enhancement Module

We conducted detailed ablation experiments on the diffusion step size and random sampling step size used in the Feature Enhancement Based on Diffusion module. First, for the

| Methods | PredCls | | SGCls | | SGDet | |
|---|---|---|---|---|---|---|
| | mR@50/100 | F@50/100 | mR@50/100 | F@50/100 | mR@50/100 | F@50/100 |
| Motifs (Zellers et al., 2018) | 16.4/17.1 | 26.2/27.2 | 8.2/8.6 | 13.2/13.8 | 6.4/7.7 | 10.5/12.5 |
| +DC (Han et al., 2022a) | 21.4/22.5 | 31.7/33.1 | 9.9/10.4 | 15.2/15.8 | 9.4/10.7 | 14.1/16.1 |
| +DHL (Zheng et al., 2023a) | 20.4/21.9 | -/- | 8.4/9.1 | -/- | 6.6/8.1 | -/- |
| +NICE (Li et al., 2022a) | 25.4/27.9 | 34.9/37.8 | -/- | -/- | -/- | -/- |
| +NICEST (Li et al., 2022a) | 24.2/26.8 | 34.16/37.23 | -/- | -/- | -/- | -/- |
| **+NoDIS** | **31.21/32.64** | **36.44/38.09** | **15.92/16.65** | **19.08/19.90** | **13.76/15.95** | **17.23/19.81** |
| VCTree (Tang et al., 2019) | 16.6/17.4 | 26.3/27.5 | 7.9/8.3 | 12.8/13.4 | 6.5/7.4 | 10.6/12.0 |
| +HTCL (Wang et al., 2023b) | 32.6/33.9 | 41.4/42.9 | 15.7/16.3 | 20.3/21.0 | 14.0/15.8 | 17.3/19.6 |
| +GCL (Dong et al., 2022) | 35.4/36.7 | 39.5/41.1 | 17.3/18.0 | 20.0/20.8 | 15.6/17.8 | 16.5/19.1 |
| VTransE (Zhang et al., 2017) | 14.0/15.0 | 22.4/23.8 | 8.1/8.7 | 13.0/13.9 | 5.8/6.6 | 9.6/10.9 |
| +GCL (Dong et al., 2022) | 30.4/32.3 | 32.8/34.7 | 16.6/17.4 | 19.2/20.0 | 14.7/16.4 | 15.0/17.2 |
| Transformer (Vaswani et al., 2017) | 19.1/20.2 | 29.5/31.1 | 9.3/9.7 | 14.6/15.2 | 6.7/7.9 | 10.7/12.6 |
| +DHL (Zheng et al., 2023a) | 18.2/20.1 | -/- | 8.7/9.3 | -/- | 7.8/8.8 | -/- |
| **+NoDIS** | **31.50/33.44** | **35.91/37.92** | **17.23/17.83** | **20.33/21.05** | **14.28/16.23** | **16.73/18.98** |

*Table 7.* Comparison with Existing Methods on Three Tasks of the GQA (Hudson & Manning, 2019a) Dataset.

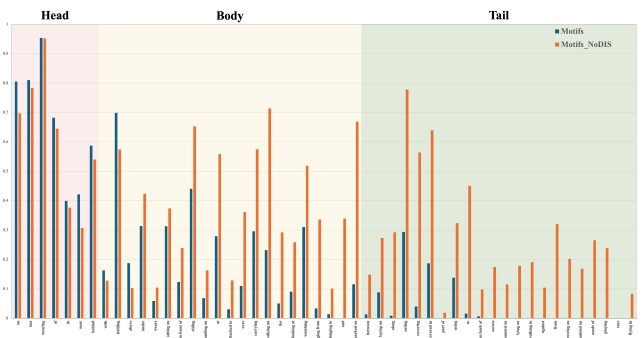

(a) Recall values for each category with Motifs and after incorporating NoDIS.

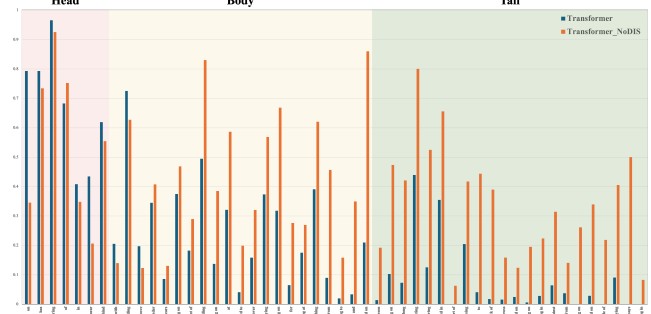

(b) Recall values for each category with Transformer and after incorporating NoDIS.

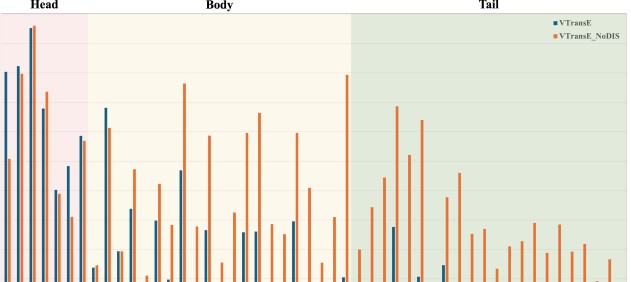

(c) Recall values for each category with VTransE and after incorporating NoDIS.

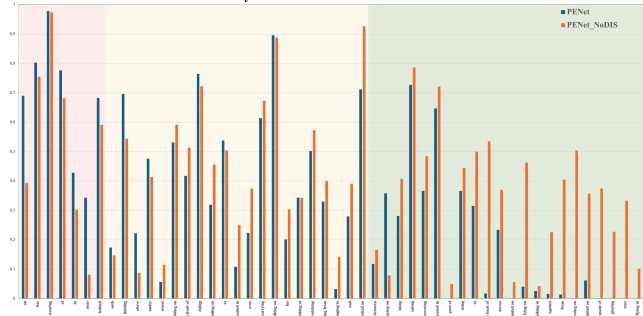

(d) Recall values for each category with PENet and after incorporating NoDIS.

*Figure 6.* Recall values for each category on the test dataset before and after incorporating NoDIS, with predicate frequencies decreasing from left to right in the dataset.

diffusion process, as shown in Table 10, we experimented with diffusion steps of 50, 100, and 150. Considering the relatively small size of our dataset, we did not adopt the commonly used 1000-step setting in traditional diffusion models (Song et al., 2020a; Chen et al., 2023). During training, a diffusion step is randomly selected from the specified range for step embedding. The results show that using 50 diffusion steps achieves the best performance in terms of mR@50 and mR@100. Therefore, we adopt 50 steps in all subsequent experiments.

Second, regarding the sampling process in this module, we conducted ablation studies on four different random sampling step sizes. As the sampling step increases, both train-

| NPR | PPA | R@50/mR@50 | R@100/mR@100 |
|-----|-----|------------|--------------|
| ✗ | ✗ | 52.92/32.53 | 55.66/35.16 |
| ✓ | ✗ | **59.89**/31.76 | **62.32**/34.06 |
| ✗ | ✓ | 47.42/33.19 | 49.61/36.16 |
| ✓ | ✓ | 54.26/**35.76** | 56.63/**38.41** |

*Table 8.* Ablation Study on Submodules of the Noise-Guided Predicate Representation Extraction Module. NPR refers to Noise-Guided Predicate Refinement, and PPA denotes the Predicate-Prototype Align module.

| $L_{ace}$ | $L_{kl}$ | R@50/mR@50 | R@100/mR@100 |
|-----------|----------|------------|--------------|
| ✗ | ✗ | 59.46/31.85 | 61.89/34.22 |
| ✓ | ✗ | 59.18/33.35 | 61.60/35.98 |
| ✗ | ✓ | **59.84**/32.15 | **62.24**/34.40 |
| ✓ | ✓ | 54.26/**35.76** | 56.63/**38.41** |

*Table 9.* Ablation Study on the Loss Functions in the Noise-Guided Predicate Representation Extraction Module.

ing and inference time tend to increase to varying degrees. As shown in Table 11, when the sampling step is set to 10, the model achieves favorable performance across multiple metrics, while maintaining reasonable training and testing times. This setting effectively balances performance and efficiency.

| diffusion steps | R@50/mR@50 | R@100/mR@100 |
|-----------------|------------|--------------|
| 50 | 49.96/**37.25** | **53.21/39.97** |
| 100 | **50.28**/36.86 | 52.74/39.54 |
| 150 | 49.73/36.90 | 52.18/39.76 |

*Table 10.* In the Feature Enhancement based on Diffusion module, an ablation study is conducted on the number of diffusion steps. During the diffusion process, step lengths are randomly selected based on the specified number of steps.

## C.3. Ablation Study on Feature Reconstruction Enhancement and Discretization Mapping Module

We conducted a detailed ablation analysis of the Feature Reconstruction Enhancement Based on Diffusion module and the Feature Discretization Mapping module. As shown in Table 12, incorporating only the Feature Discretization Mapping module or the Feature Reconstruction Enhancement Based on Diffusion module leads to a performance drop. This decline is attributed to the introduction of additional learnable variable parameters, which fail to converge effectively during training. When using the diffusion model for feature reconstruction enhancement, we introduced a random timestep reconstruction constraint to further regulate the reconstruction process, resulting in a slight performance improvement. However, a significant performance boost was observed when combining the diffusion method with

| random steps | R@50/mR@50 | R@100/mR@100 |
|--------------|------------|--------------|
| 10 | **49.96/37.25** | **53.21**/39.97 |
| 15 | 47.51/36.81 | 50.14/39.88 |
| 20 | 47.63/36.92 | 50.26/**39.99** |
| 25 | 47.54/36.85 | 50.15/39.92 |

*Table 11.* The ablation study further explores the use of randomly sampled step lengths to constrain the diffusion process. Following the DDIM-inspired (Song et al., 2020a) design, diffusion steps are randomly selected for feature reconstruction.

the discretization mapping approach. With the introduction of a conditional Gaussian noise generator, an additional diffusion constraint is applied, improving the performance to 39.66%. The aforementioned discretization mapper, as a preprocessing module for the Diffusion reconstruction task, is designed to perform feature mapping on preprocessed predicate contexts. When an additional discrete mapping is applied to the representations reconstructed by Diffusion, the model achieves optimal performance.

From the experimental analysis above, we conclude that the performance gain of a simple Diffusion model on feature reconstruction is not significant. This is because the reconstructed predicate representations are unstable and exhibit randomness, which negatively impacts subsequent predicate category predictions. However, introducing additional constraints into the diffusion process effectively enhances the quality of feature reconstruction, aligning it more closely with the expected feature distribution. Furthermore, directly using the reconstructed predicate representations for prototype classification can result in instability, as reconstructed predicate representations within the same category may fail to converge, hindering prototype optimization. By incorporating discretization mapping, the alignment between the reconstructed representations and the prototype representation space is improved, facilitating stable category predictions.

| FED | Random | Condition | CGNG | FDM | R@50/mR@50 | R@100/mR@100 |
|-----|--------|-----------|------|-----|------------|--------------|
| ✗ | ✗ | ✗ | ✗ | ▲ | 42.08/34.75 | 45.34/37.99 |
| ✓ | ✗ | ✗ | ✗ | ✗ | **54.23**/35.75 | **56.60**/38.42 |
| ✓ | ✓ | ✗ | ✗ | ✗ | 45.11/36.04 | 47.95/38.95 |
| ✓ | ✓ | ✗ | ✗ | ▲ | 45.34/36.03 | 48.13/39.18 |
| ✓ | ✓ | ✓ | ✗ | ▲ | 45.51/36.18 | 48.31/39.27 |
| ✓ | ✓ | ✓ | ✓ | ▲ | 49.98/**36.90** | 52.44/39.66 |
| ✓ | ✓ | ✓ | ✓ | ▼ | 49.72/36.85 | 52.21/**39.75** |

*Table 12.* Ablation Study of the Feature Enhancement Based on Diffusion (FED) Module and Feature Discretization Mapping (FDM) Module. "Random" indicates feature reconstruction with random time steps, "Condition" introduces the conditional diffusion method, ▲ denotes providing prior conditions for Diffusion, and ▼ indicates discretizing the features after Diffusion-based reconstruction.

## D. Quantitative and Qualitative Analysis

**Quantitative Analysis** First, we perform a visual analysis to evaluate the impact of each module in NoDIS on the intra-class variance of predicate representations. As shown in Figures 7a and 7c, the three curves represent the intra-class variance changes of predicates representation within the same class after introducing the Feature Enhancement Based on Diffusion Module and the Feature Discretization Mapping Module. After introducing the Feature Enhancement Based on Diffusion Module, we observe a significant increase in intra-class variance across all categories, with an approximately threefold improvement. This indicates that applying the diffusion-based feature enhancement and expanding the visible predicate representation space effectively enhances the diversity of within-class predicate representations, aiding the model in learning more comprehensive representation information. After introducing the Feature Discretization Mapping module for feature aggregation, the intra-class variance significantly decreases, even approaching zero. This is because, by applying the discretized feature mapping, we aggregate all scattered features based on their categories. By seeking a unified predicate representation, we group the same- category predicate representations into a single representation space, effectively controlling intra-class variance and ensuring the consistency of intra-class representations. The feature aggregation process enhances the decision layer's ability to learn and capture key information about the uniqueness of predicates within the same category. As shown in Figures 7b and 7b, we calculate the variance between the input features to the decision layer and the classification weights at the decision layer. After feature aggregation through discretized mapping, we observe that the aggregated features closely resemble the weight distribution of the corresponding category in the decision layer, with a significant reduction in variance. This demonstrates that the aggregation of features strengthens the effective representation of predicates within the same category, which in turn adjusts the decision layer's classification weights and alleviates decision-making difficulty and confusion.

Additionally, we perform a visual analysis of the aggregation of predicate representations within the same category in the Feature Discretization Mapping module. In this module, We design a learnable discretization mapping encoder that maps predicate features of the same category into a unified space by calculating the distance between these features and the encoder's representations. Figure 8a shows the distance relationships between 50 randomly selected samples and the encoder's representation space. The closer the distance, the higher the score, and the darker the color. Figure 8b illustrates the predicate discrete encoder representation assigned to each sample. By comparing Figures 8a and 8b, it is evident that after discretization mapping, each sample is distinctly assigned to the corresponding predicate category

representation space. This process effectively aggregates the diversity of predicates within the same category into a unified space, enhancing the consistency of predicate representations within the same category. It also demonstrates that our discretization encoder can effectively learn valid predicate representations for each category.

**Qualitative Analysis** We conducted a visual qualitative assessment of the model's prediction results on the PredCls task using the test dataset. As shown in Figure 9, we performed a comparative analysis between the baseline model and the predictions obtained after incorporating our method. Taking the third row of Figure 9 as an example, the baseline model (PENet(Zheng et al., 2023b), Transformer(Vaswani et al., 2017)) predicts the relationship between "tree" and "hill" as "on." While "on" is a valid predicate between these two objects, it lacks semantic richness as a head predicate category and fails to provide deeper insight into their relationship. In contrast, our method successfully generates more specific and accurate tail predicates, such as "**growing on**" and "**covered in**", with confidence scores exceeding 90%.

Additionally, in examples like the first and second rows of Figure 9, such as "bus **parked on** street" and "wheel **attached to** train," our method produces more precise and semantically informative tail predicates for a variety of object pairs. It also assigns higher confidence scores to these predicates. These results demonstrate that our method effectively understands and identifies the relational states between objects. Instead of being dominated by generalized head predicates, it assigns object pairs with tail predicates that convey more precise semantic information.

## E. Limitation Analysis

This paper introduces the Noise-Guided Predicate Representation Extraction and Diffusion-Enhanced Discretization (NoDIS) learning method, which is the first to apply diffusion models for feature enhancement to mitigate the biased prediction problem caused by long-tail distributions. However, our experiments reveal that while diffusion models exhibit strong generative capabilities, they are somewhat uncontrollable in generating results, especially when used for feature enhancement. As a result, relying solely on diffusion models for feature enhancement yields minimal improvement. To achieve more consistent representations, additional constraints must be incorporated into the diffusion process to regulate the distribution of generated features. However, introducing more constraints significantly weakens the diffusion model's generalization ability and generative performance. Therefore, designing a diffusion process for feature-level enhancement that retains the robustness and generalization capacity of the diffusion model

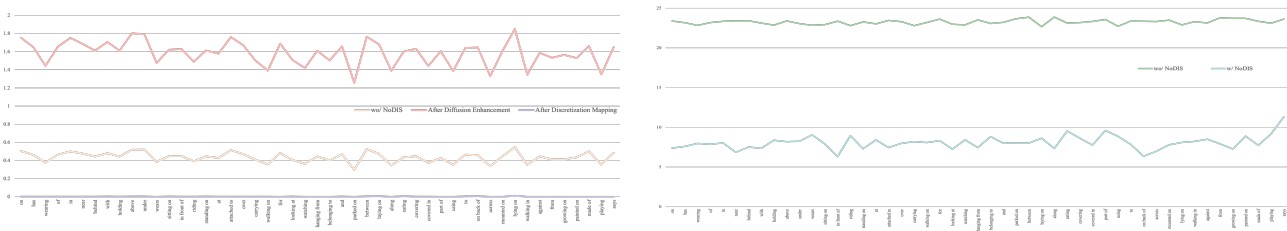

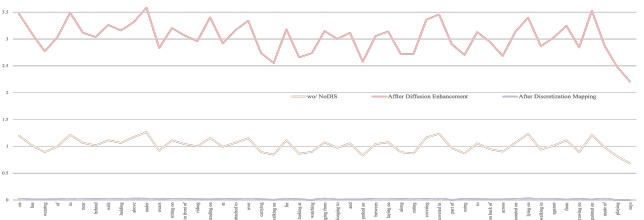

(a) Intra-class variance variation curves of PENet before and after introducing the Feature Enhancement Based on Diffusion Module and Feature Discretization Mapping module.

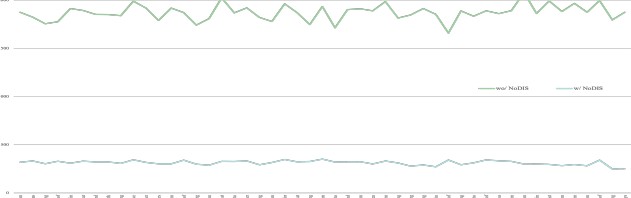

(b) Variance curves between predicate representations within the same category and their corresponding decision layer weight distributions in PENet after integrating the NoDIS module.

(c) Intra-class variance variation curves of Transformer before and after introducing the Feature Enhancement Based on Diffusion Module and Feature Discretization Mapping module.

(d) Variance curves between predicate representations within the same category and their corresponding decision layer weight distributions in Transformer after integrating the NoDIS module.

*Figure 7.* Based on different models, the variance change curves of intra-class variance before and after introducing each module of NoDIS. A larger variance indicates a greater distributional difference between the features.

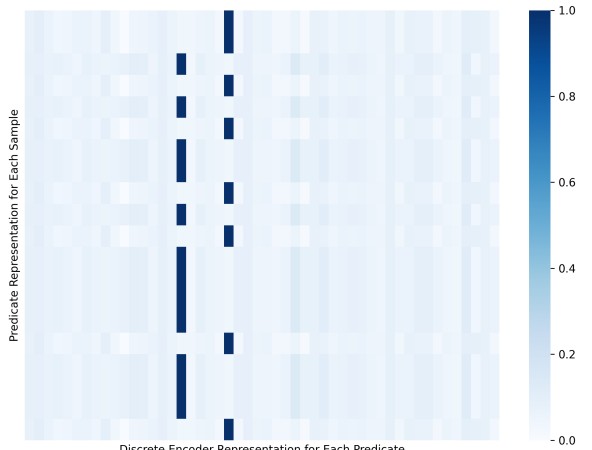

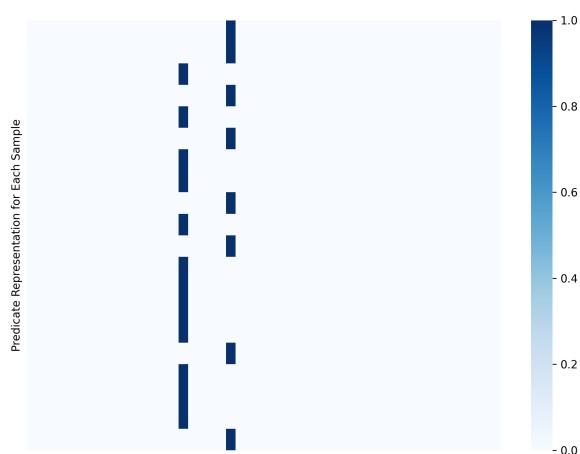

(a) Score Determination Between Similar Samples When Aggregating Predicate Features via Discretized Feature Mapping.

(b) Sample Indexing for Aggregating Predicate Representations of the Same Category via Discretized Feature Mapping.

*Figure 8.* Visualization and Evaluation of Distance Metric Scores for Aggregation of Same-Class Predicate Representations in Discretized Feature Mapping.

will be a key focus of our future work.

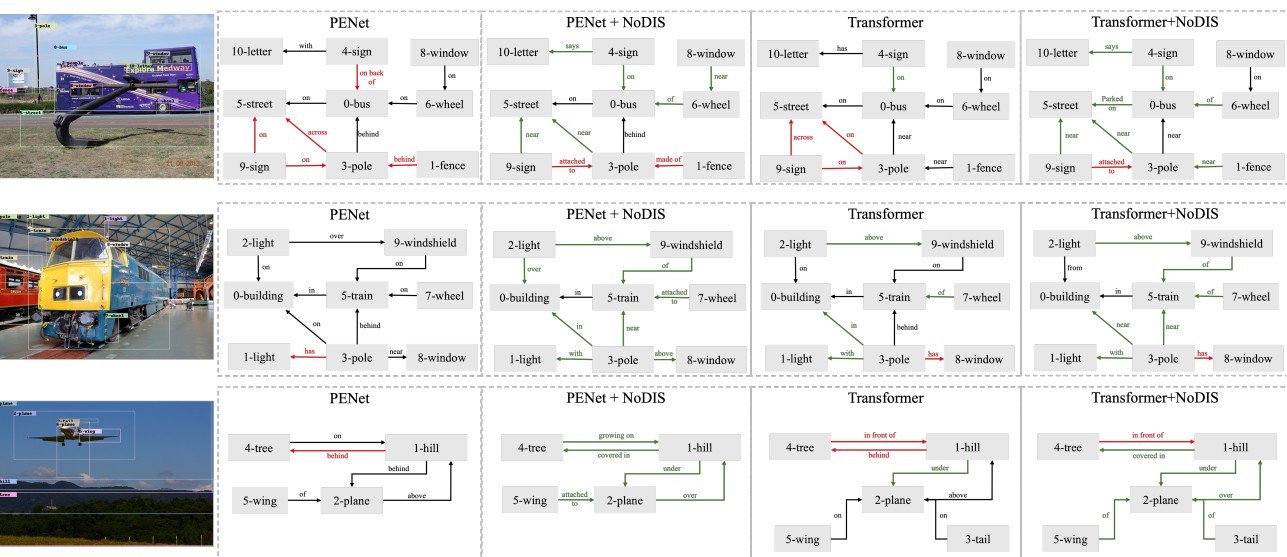

*Figure 9.* Qualitative comparisons are conducted on the test dataset based on the PredCls task. Black arrows and descriptions indicate correct predicate relationships, red ones represent incorrect predicate relationships, and green ones signify more accurate and superior predicate relationships.

