# OpenReview forum: "Noise-Guided Predicate Representation Extraction and Diffusion-Enhanced Discretization for Scene Graph Generation"
_ICML.cc/2025/Conference — ICML 2025 poster_

### Official Review · Reviewer_yxqX · 2025-03-11

**Overall Recommendation:** 3

**Summary:**

This study proposes a Noise-Guided Predicate Representation Extraction and Diffusion-Enhanced Discretization (NoDIS) technique to solve the long-tail problem inherent in the existing dataset for learning the Scene Graph Generation (SSG) model. The main contribution is that the existing proto-type learning method used diffusion pliocytosis as a method for expanding the expression space of the predicate feature. Additionally, feature gathers as a decision boundary that can occur as the representation space is determined, and a discretization mapper is introduced to prevent inference performance degradation. This study was verified on the benchmark for SGG and proved effective in alleviating the long-tail problem.

**Claims And Evidence:**

- In this paper, the proposed NoDIS can improve prediction of feature. This is consistent with my intuition, but I have questions about whether it can be argued in general. It can be said that the diversity of feature representation may increase in the process of adding noise and de-noise during the diffusion process, but there is a lack of clear evidence (the effect of NoDIS can be confirmed by the ablation of Tab. 2 in this text, but I think it is too fragmentary evidence). As in the PE-Net study used as a baseline in this paper, if the results of feature representation through t-SNE are attached, the trust in the argument will be improved.
- The Learnable Feature Discretization Mapping Module proposed in Section 3.4 is shown to avoid confusion between inter-classes that may occur during the diffusion process. If so, it is thought that the change in feature distance between inter-classes is also necessary, as reported in Figure 1 (b) of the text for the change in variance between the intra-classes. If the module worked effectively, the feature distance between the inter-classes would increase further.

**Essential References Not Discussed:**

- There is a missing one of the latest papers dealing with the long-tail issue based on two-stage, the same as in this study. Please refer to it.

Kim, Hyeongjin, et al. "Scene graph generation strategy with co-occurrence knowledge and learnable term frequency." Proceedings of the 41st International Conference on Machine Learning. 2024.

**Experimental Designs Or Analyses:**

- The Visual Genome and GQA datasets used in the experiment are used as major data sets for SGG performance evaluation, and there seems to be no problem in selection.
- Models set as performance comparators are two-stage-based models and are suitable as performance comparators.
- GQA is mentioned as the usage dataset, but not found in the actual experimental table. Without experiments on additional benchmark datasets, the claims of model's general performance improvement are highly problematic.
- It is reasonable to report the recall values of each head, body, and tail class to prove the Long-tail problem mitigation, but the comparison group is not appropriate. For the proposed effect of NoDIS, it is considered correct that baseline, baseline + prototype, baseline + NoDIS, and baseline + all are used to prove the effectiveness.
- The above problem is also shown in Tab. 2 of ablation study, which shows better performance when using only prototype than when using NoDIS only. This can be judged as the main contribution of the performance increase by the prototype learning method rather than the proposed NoDIS.
- Tab. 6, 7 in Appendix is the same experiment as in text Tab. 2, 3. If it's unnecessary, it's right to remove it.

**Methods And Evaluation Criteria:**

- The proposed model was evaluated on the existing two-stage-based SGG model, which can be seen as fair progress in terms of performance comparison with the existing two-state model.
- The benchmark data used in the experiment, Visual Genome, and GQA datasets, are used as major data for SGG performance evaluation, which can be considered valid as data sets for performance comparison experiments.
- All three subtasks (PredCls, SGCls, SGDet) performances that are key to SGG performance comparison are reported, and R@100 values per head, body, and tail class are well reported for proof of long-tail problem mitigation.
- The Implementation Detail aspect for reproducibility of experimental performance is somewhat inadequate. Values such as learning rate scheduler and feature dimension used in the experiment are missing.
- It seems that the experimental results on the GQA dataset are missing. The paper says that it was used as performance evaluation data in sec. 4.1, but we can't find the experimental results in either the main or the supplement.

**Other Comments Or Suggestions:**

- Many figure diagrams are missing descriptions of x-axis and y-axis. There is an inconvenience in interpreting the diagram.
- The formula is well written, but it is difficult to follow because there are many identical expressions. For example, in the case of Eq. 5 , it is difficult to distinguish between pre- and post-operation T.
- The same expression in Eq. 10 is ambiguous. My personal understanding is that you only use the Q_p vectors that you have concatenated when constructing T, but we need to use a better expression. Mathematically, it seems that we only use the first element.
- The letter and arrow are overlaid at the top (c) of Figure 2, which needs to be corrected.
- Tab. 2 looks better to replace with Tab. 6 from Appendix. Both experiments are the same, and for Tab. 6, both NPR and PPA are reported to have unused baseline performance, making it easier for experimental evaluation.
- Unification is necessary when referring to Figure in the text. Some are denoted as Figure xb, while others are denoted as Figure x(b).

**Other Strengths And Weaknesses:**

Strength
- The introduction of a diffusion policy for solving the long-tail problem, which is the challenge problem of the existing scene graph dataset, is evaluated creatively.
- The paper explains the proposed method in detail, and the formula and algorithm table are well organized.

Weakness
- Experiments to prove effectiveness on the proposed NoDIS are somewhat lacking
- Lack of related work content. In order to claim the improvement of feature presentation through the diffusion policy claimed in this paper, research related to this must be added.
- It feels somewhat old compared to open-vocabulary studies using the latest one-stage or LLM.

**Questions For Authors:**

- To verify the effectiveness of the proposed NoDIS, experiments on the change of feature distance between t-SNE and inter-classes such as in PE-Net are needed.
- What is the basis for improving feature representation when using the Diffusion policy? If you have that study, it is recommended to add it to the related work. This enhances the reader's understanding of the thesis.
- Is the experiment on GQA missing? As far as I've reviewed, it's nowhere to be seen in the text and Appendix. This takes a toll on the suggestion method generalization argument.
- The latest SGG research trend is dominated by the one-stage method, and I wonder why you adopted the two-stage method. Is there any particular reason?

**Relation To Broader Scientific Literature:**

- Although the research on Scene graph is well organized, the general content of Diffusion is far from improving feature representation in the case of Diffusion studies. To reinforce the research claim, the reinforcement of feature presentation through the Diffusion policy should be added to the relatiob work.
- The proposed paper focused on two-stage-based SGG models, but recent research trends have mainly focused on one-stage, which can train SGG models end-to-end. In addition, with the development of the Large Language Model (LLM), this study feels somewhat old compared to the study dealing with long-tail research through open-vocabulary with LLM.

**Theoretical Claims:**

- This paper was not accompanied by any specific mathematical proof.
- Although the paper argues for feature representation enhancement through diffusion policy, the theoretical basis for this is lacking.
- All proofs of the claims have been performed as inductive proofs, but I believe that additional strong experimental evidence of the claims is needed.

---

> ### Author Rebuttal · Authors · 2025-03-31
>
> ## We sincerely appreciate your time and effort in carefully reviewing our paper and providing constructive feedback. We are also grateful for your recognition of our work.
> - *For Weakness 1*: Thank you for bringing this up. First, we compared and analyzed our method against state-of-the-art approaches using the same evaluation framework on the VG dataset (in Section 4.2 and Table 1). Additionally, **we conducted comparisons on the Open Images V4, V6 (results in Reviewer giRZ), and GQA (results in Reviewer FDij) datasets**. Finally, we performed an ablation study by breaking down our method and analyzing its effectiveness from multiple perspectives, **including inter-module effectiveness (in Table 2), the impact of the diffusion method (in Table 4), loss function design (in Table 3), diffusion step size (in Reviewer FDij), and denoising step size (in Reviewer FDij)**.
> - *For Weakness 2*: Thank you for pointing this out. Current diffusion-based methods are primarily used for generation tasks. Some studies also apply diffusion for data augmentation by generating additional pseudo-samples to enhance tail information and improve model performance. We have included this discussion in the related work section of our paper.
> - *For Weakness 3*: Thank you for pointing this out. **Although existing LLMs demonstrate strong image understanding capabilities, they widely suffer from hallucination issues, largely due to data bias**. For example, when asked about the relationship between "person" and "short" in an image, LLaVA incorrectly responds: "The person is standing next to a short." However, when guided by NoDIS, LLaVA correctly outputs: "The person is wearing shorts." We will include more details and additional results in the appendix. Therefore, mitigating biased outputs caused by data bias remains a critical challenge. Moreover, while LVLMs have extremely large parameter sizes, NoDIS contains only around 300M parameters (in reviewer giRZ), requiring significantly less GPU memory for training. Optimizing and experimenting with smaller models can better support large models and help alleviate biased predictions.
> - *For Question 1*: Thank you for pointing that out. We previously used t-SNE for visualizing the representation space. However, we considered that t-SNE itself may have some randomness (as it requires dimensionality reduction). Additionally, after feature discretization mapping, our method clusters multiple samples of similar predicates into a unified representation space. As a result, visually, it appears as a distinct small region, whereas PE-Net appears as a more scattered large region, which does not look aesthetically pleasing. Therefore, instead of using t-SNE for visualization, we calculated inter-feature relationships and presented the results numerically. We chose not to include the t-SNE results in the paper, but we can certainly add them in the appendix of the final paper.
> - *For Question 2*: Thank you for pointing this out. In SGG tasks, no existing work has utilized Diffusion models to enhance features for expanding the tail representation space. Instead, most approaches rely on pretrained Diffusion or GAN models for data augmentation, which incurs higher computational costs. Some methods expand feature representations by training a decoder and applying a distillation-like approach, as briefly mentioned in the Introduction. We will include a detailed discussion of these methods in the Related Work section.
> - *For Question 3*: Apologies for the omission due to space constraints. **We not only evaluated our method on GQA but also conducted experiments on Open Images V4 and V6**. **The GQA results can be found in our response to Reviewer FDij**, while **the Open Images V4 and V6 results are provided in our response to Reviewer giRZ**. These results demonstrate that our method achieves outstanding performance across multiple datasets, highlighting its strong generalization capability.
> - *For Question 4*: Thank you for pointing this out. Two-stage methods primarily rely on Faster R-CNN's entity detection capabilities to extract entity representations and construct relationship information by independently training a relationship detection module. In contrast, single-stage frameworks based on the Transformer architecture perform object detection and relationship classification simultaneously. While single-stage methods are simpler, they require higher training costs. Moreover, existing single-stage methods are relatively scarce and generally underperform compared to two-stage methods, as shown in the table.
> |Method|Stage|mR@50|mR@100|
> |:-------:|:-------:|:-------:|:-------:|
> |RelTR|One-stage|10.8|-|
> |SSR-CNN|One-stage|8.4|10.0|
> |Iterative SGG|One-stage|8.0|8.8|
> |Relationformer|One-stage|9.3|10.7|
> |SGTR|One-stage|12.0|15.2|
> |**Transformer-NoDIS(ours)**|two-stage|**16.91**|**19.25**|
>
> ### If you have any further questions or concerns, please let us know, and we will provide additional clarification.

---

> > ### Comment · Reviewer_yxqX · 2025-04-02
> >
> > Overall, the authors have responded to all reviewer comments with logical consistency and detailed justifications supported by quantitative evidence. In particular, they have clearly expressed their intention to address the identified experimental and theoretical limitations, and have also proposed to include additional references and experimental results. Therefore, the response is considered to be both sincere and sufficiently thorough. Based on this, I will consider to rise current score.

---

> > > ### Author Response · Authors · 2025-04-06
> > >
> > > Thank you very much for your recognition of our work, and thank you very much for improving the score!

---

### Official Review · Reviewer_FDij · 2025-03-12

**Overall Recommendation:** 3

**Summary:**

The paper addresses bias in scene graph generation, especially the difficulty of learning fine-grained predicate labels under long-tailed distributions. The authors propose NoDIS, which has two core aspects: First, Noise-Guided Predicate Extraction expands predicate representations by applying a single-step noise addition and iterative denoising, then further augments this diversity through a conditional diffusion mechanism. This helps generate richer, more diverse representations for both frequent (head) and infrequent (tail) predicates. Second, Feature Discretization Mapping learns to aggregate these expanded predicate features into a discretized representation. This step reduces ambiguity for the classification head by unifying predicate samples of the same category into a more consistent cluster. Empirical evaluations on the Visual Genome and GQA benchmarks demonstrate that adding NoDIS boosts the performance on rare classes. Ablation studies confirm that both the diffusion-based augmentation and the discretization module are integral to the performance gains.

**Claims And Evidence:**

1. The paper states that existing SGG approaches overlook intraclass diversity and interclass consistency in predicate representations, leading to biased predictions favoring head classes. They show that prior efforts primarily focus on loss reweighting or sampling strategies, and only partially address the wide range of predicate semantics. Quantitative results (particularly in mR@K metrics) show that standard baselines often fail to predict tail predicates accurately.

2. The proposed NoDIS method, by applying noise-guided extraction and a conditional diffusion model, effectively expands the representation space of predicates, improving the recall on long-tailed predicates. Comparisons on the VG dataset indicate that the average performance on tail categories (mR@K) is consistently higher compared to baselines. Their ablation study further isolates the impact of the diffusion approach on increasing within-class variance of predicate features.

3. The discretization mapper helps unify predicate features that belong to the same category, improving classification accuracy and reducing confusion among semantically similar predicates. The authors show a measurable drop in intraclass feature variance once discretization is applied and an alignment between the learned representation distributions and the final decision layer. This is reinforced by the improvement in F@K and mR@K metrics.

While the paper does not provide an in-depth theoretical proof that this particular combination of modules is uniquely optimal, their experimental findings and ablations make a convincing empirical case that NoDIS alleviates biases and achieves strong performance on standard SGG benchmarks.

**Essential References Not Discussed:**

The references to mainstream SGG and diffusion-based generation are mostly in place.

**Experimental Designs Or Analyses:**

The authors compare performance on three tasks (PredCls, SGCls, SGDet) using widely adopted metrics (R@K, mR@K, F@K).
All in all, the experimental protocols appear sound and consistent with common SGG benchmarks.
One area for improvement might be clarifying whether their random noise schedules significantly impact results or whether different hyperparameter choices matter (e.g., number of diffusion steps).

**Methods And Evaluation Criteria:**

The authors propose two-phase design: (1) object detection, (2) predicate classification enhanced by noise-guided extraction and conditional diffusion. Noise injection and denoising are used to create more distinct predicate embeddings (intra-class diversity). A discretization mapping collapses these diverse embeddings for each category, reducing confusion at the classifier stage. The chosen evaluations are well aligned: the authors address data skew and demonstrate improvements on established unbiased metrics (mR@K, F@K).

**Other Comments Or Suggestions:**

- Experimental table (Table 1) might benefit from consistent significant figures of results.
- If possible, consider making the figure for NoDIS more straightforward by reducing the complexity of parallel branches or clarifying the ordering of steps in the visual layout.

**Other Strengths And Weaknesses:**

Strengths:
1. The conceptual combination of diffusion-based expansion plus discretized representation is novel, improving both the expressiveness (helpful for tail classes) and consistency (for simpler classification).
2. Improved tail performance is clearly demonstrated by the metric gains on mR@K.

Weaknesses:
1. Figure 2(b) is indeed somewhat hard to parse, due to the overlapping gray/dotted arrows and the large number of parallel lines (the textual explanation helps, but the graphic is busy).
2. Hyperparameter sensitivities (e.g., noise schedule, number of diffusion steps, discretization granularity) are not deeply explored. In practice, these can matter greatly for diffusion-based methods.
3. The inclusion of more recent baselines would strengthen the comparisons and provide a more comprehensive evaluation.

**Questions For Authors:**

1. How sensitive are results to different noise schedules or varying the number of denoising steps?
2. At inference, once you generate expanded predicate features, are they discretized solely by a nearest-neighbor approach to a learned embedding codebook? Would an alternative approach (e.g., approximate nearest neighbors or distinct heads for each class) help or hinder the method?
3. Have you considered deeper comparisons or breakdowns of tail performance specifically on GQA? Because GQA has many predicate categories, this might better showcase the benefit of diffusion-based augmentation.

**Relation To Broader Scientific Literature:**

- Long-tailed recognition: Instead of conventional class-rebalancing or cost-sensitive methods, they utilize a generative model (diffusion) to expand underrepresented classes.
- SGG bias reduction: Past works have used sampling, reweighting, or knowledge distillation. NoDIS is somewhat unique in employing a diffusion-based pipeline to construct richer embeddings.
- Feature discretization: The idea of quantizing or discretizing features with a VQ-VAE–style approach to unify sample clusters seems also a novel adaptation in SGG.

**Theoretical Claims:**

The paper’s theoretical contributions are mostly method-centric rather than purely formal proofs. The authors do mention a KL-divergence loss to align representation distributions and reference how the diffusion model preserves or shifts distributions across timesteps. While they do not include a formal proof of correctness or convergence for the diffusion approach, the claims rely on standard diffusion modeling assumptions. I did not see any outright flaws in their theoretical arguments, though the proofs and formal rigor focus on standard diffusion derivations and established VAE/GAN-based feature augmentation concepts rather than novel mathematics.

---

> ### Author Rebuttal · Authors · 2025-03-31
>
> ## We sincerely appreciate your time and effort in carefully reviewing our paper and providing constructive feedback. We are also grateful for your recognition of our work.
> - *For Weakness 1*: Thank you for bringing this up. Figure 2(b) illustrates both the training and inference processes, making it relatively complex. We have now optimized the diagram by simplifying redundant module structures, resulting in a clearer and more concise representation.
> - *For Weakness2*: Thank you for pointing this out. We follow the existing noise design of SD and DiT, adopting a linear noise schedule. To further investigate the impact of diffusion steps on performance, we conduct an ablation study with 50, 100, and 150 steps, as shown in Table 1. Additionally, to examine the effect of step size during the denoising process, we perform another ablation study with step sizes of 10, 15, 20, and 25, as presented in Table 2. A more detailed ablation analysis will be provided in the final paper.
> |Diffusion steps|random steps| R@50/mR@50 | R@100/mR@100 | F@50/F@100 |
> |:------:|:------:|:------:|:------:|:------:|
> | 50 | 10 | 49.96/**37.25**  | **53.21**/**39.97**  | **42.68**/**45.65** |
> | 100 | 10 | **50.28**/36.86  | 52.74/39.54 | 42.54/45.20 |
> | 150 | 10 |  49.73/36.90 | 52.18/39.76  | 42.36/45.13 |
>
>     |random steps|R@50/mR@50|R@100/mR@100|F@50/F@100|
>     |:------:|:------:|:------:|:------:|
>     |10|**49.96**/**37.25**|**53.21**/39.97|**42.68**/**45.65**|
>     |15 | 47.51/36.81| 50.14/39.88  | 41.48/44.42 |
>     |20 | 47.63/36.92| 50.26/**39.99**  | 41.60/44.54 |
>     |25 | 47.54/36.85|50.15/39.92|41.52/44.46|
> - *For Weakness 3*: Thank you for your suggestion. On the VG dataset for the PredCls task, using Transformer as the baseline, our method outperforms the approach published in TMM 2024 by 0.35 and the one in TIP 2024 by 0.95 in F@100. In terms of overall performance, with PE-Net as the baseline, our method surpasses the CVPR 2024 approach by 0.73 in mR@100. Additionally, on the Open Images dataset, based on the score_wtd metric (as referenced in Reviewer giRZ's response), our method improves upon the best existing approach by 0.29 on OI V4 and by 1.05 on OI V6.
> - *For Suggestions*: Thank you for your suggestion. We have optimized the content accordingly and will present it in the final paper. We sincerely appreciate your thorough review of our paper!
> - *For Questions 1*: Thank you for pointing this out. In Section 4.3 and Table 4 of the paper, we briefly analyze the necessity of incorporating a random denoising strategy for supervised optimization. During the forward diffusion process, **our noise scheduling strategy follows that of SD and DiT**. To better enforce consistency in the generated representations of the enhanced model, we introduce an additional denoising strategy by randomly sampling n steps for denoising as an extra constraint. During inference, we adopt the DDPM denoising strategy to iteratively remove noise step by step. **Experimental results on different diffusion and denoising step selections are presented in For Weakness2**. We will include this analysis in the final version of the paper with a more detailed discussion.
> - *For Questions 2*: Thank you for pointing this out. Our initial approach considered using the K-nearest neighbors (KNN) method. However, since Diffusion generates diverse features in the early stages, predicate representations within the same category exhibit significant variation, making it ineffective to constrain them using KNN. Instead, we maintain a component similar to a CodeBook to learn a central representation from the representation space of multiple samples within the same category. This central representation is designed to have strong generalization capabilities. In contrast, KNN and clustering methods are too restrictive and struggle to model diverse representations effectively.
> - *For Questions 3*: Thank you for pointing this out, and we apologize for not including it in the paper due to space limitations. The table below compares our method with existing approaches on the PredCLS task of the GQA dataset. Our method achieves further performance improvements over existing methods. More comprehensive evaluation results across multiple tasks will be included in the final version of the paper. Additionally, we have supplemented performance improvements on the Open Images V4 and V6 datasets (results in Reviewer giRZ), which will also be included in the final version.
> | Method  | R@50 | mR@50  | F@50  | R@100 | mR@100  | F@100  |
> |:------:|:------:|:------:|:------:|:------:|:------:|:------:|
> | Motifs | **65.3** | 16.4 | 26.2 | **66.8** | 17.1 | 27.2 |
> | Motifs-DC |61.3|21.4|31.7|62.7|22.5|33.1|
> | Transformer |65.2|19.1|29.5|66.7|20.2|31.1|
> | Transformer-DHL|-|18.2|-|-|20.1|-|
> | **Transformer-NoDIS(Ours)**|46.54|**30.47**|**36.83**|48.45|**31.90**|**38.47**|
>
> ### If you have any further questions or concerns, please let us know, and we will provide additional clarification.

---

> > ### Comment · Reviewer_FDij · 2025-04-07
> >
> > Thank you for preparing a detailed rebuttal. I do find the work overall interesting and promising. I will be keeping my score as is.

---

> > > ### Author Response · Authors · 2025-04-09
> > >
> > > Thank you very much for your recognition of our work. We have also revised the paper according to your comments. Finally, thank you for taking the time to read our response!

---

### Official Review · Reviewer_giRZ · 2025-03-13

**Overall Recommendation:** 3

**Summary:**

The paper proposes NoDIS, a method designed to address bias in SGG arising from long-tail predicate distributions. The contributions are: 1) it introduces a noise-guided predicate representation extraction technique that employs conditional diffusion models to increase the diversity of predicate representations within the same category. 2) it develops a diffusion-enhanced discretization module to unify similar predicate representations, simplifying decision-layer complexity and improving prediction accuracy. Evaluations on VG and GQA datasets demonstrate that NoDIS outperforms existing unbias methods.

**Claims And Evidence:**

1. The authors claim improved generalization by diversifying predicate representations and reducing prediction ambiguity via discretization mapping. These claims are convincingly supported by experiments on VG and GQA datasets.
2. The quantitative analysis demonstrates improvements over baselines, especially in terms of mR@K, indicating effective bias mitigation.
3. Ablation studies effectively validate the impact of each proposed module.

**Essential References Not Discussed:**

N/A

**Experimental Designs Or Analyses:**

The experimental designs and analyses appear sound:
1. ablation studies systematically demonstrate the contributions of each module (Noise-Guided Predicate Refinement, Diffusion-Enhanced Feature Enhancement, and Discretization Mapping).
2. Comparisons with multiple state-of-the-art methods across established benchmarks validate the effectiveness of the proposed method comprehensively.

**Methods And Evaluation Criteria:**

1. standard benchmark datasets (VG and GQA) aligns well with community standards.
2. the evaluation metrics (R@K, mR@K, F@K) are conventional and sensible for measuring performance in SGG, particularly mR@K for capturing both head and tail predicate predictions.

**Other Comments Or Suggestions:**

Minor suggestions and typos:
1. Typos/Minor corrections:
 - Section 3.2: clarify more explicitly the role of the Query Token initialization.
 - Table formatting (eg, Table 4 in the ablation studies section) could be improved slightly for readability.

Suggestion:
1. Provide additional qualitative visualization examples or failure cases for more transparency

**Other Strengths And Weaknesses:**

Strengths:
1. Clearly motivated approach to tackling long-tail distributions and bias.
2. Innovative integration of diffusion models for feature diversification.
3. Strong empirical validation and ablation studies.

Weaknesses:
1. Complexity in the methodology might limit reproducibility without extensive supplementary details.
2. Limited exploration on generalizability beyond standard benchmarks, and also can explore the results in PSG and OpenImage dataset.

**Questions For Authors:**

1. Diffusion Complexity: given the computational cost associated with diffusion models, can you clarify how scalable your approach is to larger-scale datasets or real-world scenarios?
er will help assess the generalization and robustness of the method beyond benchmarks.)
2. Comparison with Other Generative Models: Why was diffusion specifically chosen for feature augmentation over alternative generative models such as GANs or VAEs? If use GANs or VAEs, what's the performance.

**Relation To Broader Scientific Literature:**

The paper situates itself within recent SGG literature, clearly discussing related prior methods like Motifs and RelTR, and identifying their limitations, especially regarding biases and handling tail classes. The paper differentiates itself by introducing novel diffusion-based methods and discretization techniques, improving over previous augmentation and feature enhancement methods.

**Theoretical Claims:**

N/A

---

> ### Author Rebuttal · Authors · 2025-03-31
>
> ## We sincerely appreciate your time and effort in carefully reviewing our paper and providing constructive feedback. We are also grateful for your recognition of our work.
> - *For Weakness 1*: Thank you very much for pointing that out. This work is the first to apply diffusion models for feature enhancement in the SGG task. To facilitate reproduction by other researchers, we have provided detailed implementation details in the appendix(Section: A. Implementation Details), including hyperparameters such as learning rate, number of iterations, and batch size. Additionally, we have submitted the original code as a supplementary file. Upon acceptance of the paper, we will upload the code to GitHub and include the repository link in the paper for broader accessibility.
> - *For Weakness 2*: Thank you for your constructive feedback. We followed the same training procedure and evaluation criteria to validate the reliability of our method on Open Images V4 and V6, as shown in the table below. Compared to recent approaches, our method demonstrates a significant advantage on the Open Images dataset, achieving the best performance in wmAP_rel, wmAP_phr, and score_wtd. This highlights our method's strong capability in both relationship detection and phrase-level consistency assessment, effectively and accurately capturing relationship details between objects. A detailed comparison and analysis will be included in the final paper.
> |Dataset|Method|mR@50|R@50|F@50|wmAP_rel|wmAP_phr|score_wtd|
> |:------:|:------:|:------:|:------:|:------:|:------:|:------:|:------:|
> |OI V4|RelDN|70.4|**75.7**|73.0|36.1|39.9|45.2|
> |OI V4|GPS-Net |69.5|74.66| 72.0 |35.0|39.4|44.7|
> |OI V4|BGNN |**72.1**|75.5|**73.8**|37.8|41.7|46.9|
> |OI V4|**Transformer-NoDIS(ours)**|70.34|74.84|72.52|**38.21**|**42.35**|**47.19**|
> |OI V6|DBiased|42.1|74.6|53.8|34.3|34.4|42.3|
> |OI V6|PENet|39.3|**76.5**|51.9|36.6|37.4|44.9|
> |OI V6|SGTR|42.6|59.9|49.8|38.7|37.0|42.3|
> |OI V6|CSL|41.7|75.4|53.7|34.3|35.4|42.9|
> |OI V6|BCTR|48.8|68.6|57.0|36.0|**39.0**|42.5|
> |OI V6|SQUAT|-|75.8|-|34.9|35.9|43.5|
> |OI V6|**Transformer-NoDIS(ours)**|**48.93**|74.11|**58.94**|**38.87**|**38.95**|**45.95**|
> - *For Minor Suggestion*: Thank you for your suggestions. In Section 3.2, we randomly initialize learnable Query Tokens based on a Gaussian distribution to extract independent predicate information from the entity representation space. By leveraging a coarse-grained denoising process, entities are treated as noise, allowing the Query Token to extract predicate representations independently of entities, thereby preventing entity influence on predicate classification. For the ablation study in Table 4, we will restructure it to analyze different aspects, including the denoising method, conditional introduction, denoising steps, and feature discretization mapping. Due to page limitations, we have simplified the overall experimental results. Additionally, we will include our failed experiments in the final paper appendix. We sincerely appreciate your thorough review of our paper!
> - *For Suggestion 1*: Thank you very much for your constructive suggestions. Due to space limitations, we have not included more failure cases and analysis. We will add additional visualizations of both successful and failed cases, along with further analysis, in the appendix. We sincerely appreciate your thorough review of our paper!
> - *For Question 1*: Thank you for your question. Our method differs from diffusion-based approaches like SD and DiT. Instead of a full diffusion model, **we adopt the diffusion concept and use only three Cross-Attention layers and three Linear layers in the noise prediction module**. The table below presents a comparison of overall model parameters. Additionally, we conducted experiments not only on the VG dataset but also on large-scale datasets such as Open Images and GQA (results in reviewer FDij). Our method outperforms existing approaches, demonstrating an effective trade-off between resource efficiency and performance while also proving its robustness.
>     |Method|Params(M)|
>     |:------:|:------:|
>     |EBM|322.2|
>     |VCTree-TDE|361.3|
>     |VCTree-EBM|372.5|
>     |BGNN | 341.9|
>     |**VCTree-NoDIS(ours)**|**306.06**|
>     |**Motifs-NoDIS(ours)**|**314.8**|
>     |**Transformer-NoDIS(ours)**|**295.42**|
> - *For Question 2*: We appreciate your question. We previously attempted online feature enhancement based on VAE and GAN, but these methods often encountered gradient explosion or vanishing issues during training. Our analysis indicates that VAE and GAN heavily rely on pre-trained weights. Moreover, **both methods(VAEs, GANs) model the overall pixel information of an image, whereas the SGG task determines relationships based on localized ROI features**. As a result, these methods fail to train effectively in this task and cannot enhance feature representation.
>
> ### If you have any further questions or concerns, please let us know, and we will provide additional clarification.

---

### Official Review · Reviewer_gBLn · 2025-03-14

**Overall Recommendation:** 3

**Summary:**

The paper introduces a diffusion-based feature enhancement approach to broaden the visual space of predicate representation and improve feature diversity. It first extracts entity representations using a baseline model and refines them with a Transformer for contextualization. Gaussian noise is then applied to the contextualized predicate features, and a diffusion-based model estimates and removes the noise. To address the discrete distribution challenge inherent in diffusion-based models, a mapper module is introduced for predicate classification. The method is evaluated with three different baselines and significantly improves F@K.

**Claims And Evidence:**

The claims made in the submission are supported by clear and convincing evidence.

**Essential References Not Discussed:**

N/A

**Experimental Designs Or Analyses:**

The experimental designs follow the standard protocol of SGG. They have demonstrated their performance with sufficient ablation studies.

**Methods And Evaluation Criteria:**

The methods and evaluation criteria is appropriate for scene graph generation.

**Other Comments Or Suggestions:**

N/A

**Other Strengths And Weaknesses:**

The paper successfully applies the diffusion-based denoising in classifying the predicates of scene graphs. The motivation to employ diffusion-based denoising is well-written and also their proposed method is discussed coherently. Proper ablation studies and evaluation with SOTA baselines demonstrate the superiority of the proposed method.

However, the paper lacks in discussing the classification of subject and object. Since they are contextualizing baseline entity features, the entity classification results will be different from the baseline models. Also, a overview of the method should be included in the caption of Figure 2.

**Questions For Authors:**

See weaknesses

**Relation To Broader Scientific Literature:**

The paper finds an interesting application of diffusion-based denoising in enhancing the visual space of predicates.

**Theoretical Claims:**

The paper does not have any theoretical claims. It essentially borrows the variants of Diffusion-based modelling into scene graph generation, especially to model the predicate representation.

---

> ### Author Rebuttal · Authors · 2025-03-31
>
> ### We sincerely appreciate your time and effort in carefully reviewing our paper and providing constructive feedback. We are also grateful for your recognition of our work.
>
> - #### *For Weaknesses*: Thank you very much for pointing this out. We classify subject and object entities using the same method as the baseline, which relies on ROI features extracted by Faster R-CNN and determines the entity category using a simple linear layer. We will update Section 3.1 of the paper to include an explanation of this part.
>
> ### If you have any further questions or concerns, please let us know, and we will provide additional clarification.

---

### Decision · Program_Chairs · 2025-05-01

**Decision:**

Accept (poster)

**Comment:**

This paper received overall positive reviews during the initial review period. The reviewers appreciated the core contributions and the novelty of the proposed approach. Some weaknesses were noted, including limited discussion of subject-object classification, missing GQA results, and suggestions to improve visualizations and explore hyperparameter sensitivity. The authors provided a rebuttal that effectively addresses many of these concerns. The AC concurs with the reviewers that this paper presents a novel approach to the SGG community and recommends acceptance. The authors are encouraged to consider the suggestions in the final version, particularly around clarifying figures and expanding results on additional datasets.